# REMINDRAG: Low-Cost LLM-Guided Knowledge Graph Traversal for Efficient RAG

**Yikuan Hu**[1◇]    **Jifeng Zhu**[1◇]    **Lanrui Tang**[2]    **Chen Huang**[13*]

[1]College of Computer Science, Sichuan University, China
[2]School of Computing and Data Science, The University of Hong Kong, Hong Kong, China
[3]Institute of Data Science, National University of Singapore, Singapore
`kilgrim@foxmail.com`   `zen1984.kr@gmail.com`
`ray8823@connect.hku.hk`   `huang_chen@nus.edu.sg`

## Abstract

Knowledge graphs (KGs), with their structured representation capabilities, offer promising avenue for enhancing Retrieval Augmented Generation (RAG) systems, leading to the development of KG-RAG systems. Nevertheless, existing methods often struggle to achieve effective synergy between system effectiveness and cost efficiency, leading to neither unsatisfying performance nor excessive LLM prompt tokens and inference time. To this end, this paper proposes REMINDRAG, which employs an LLM-guided graph traversal featuring node exploration, node exploitation, and, most notably, memory replay, to improve both system effectiveness and cost efficiency. Specifically, REMINDRAG memorizes traversal experience within KG edge embeddings, mirroring the way LLMs "memorize" world knowledge within their parameters, but in a train-free manner. We theoretically and experimentally confirm the effectiveness of REMINDRAG, demonstrating its superiority over existing baselines across various benchmark datasets and LLM backbones. Our code is available at `https://github.com/kilgrims/ReMindRAG`.

## 1 Introduction

Retrieval-Augmented Generation (RAG) [23] has become a prominent method for augmenting Large Language Models (LLMs) with external knowledge resources, enabling them to generate more accurate outputs and thereby expanding their practical utility [17, 32]. However, conventional RAG approaches, which primarily employ dense vector retrieval [37, 20, 1] for identifying relevant text segments, often exhibit limitations when confronted with intricate tasks that necessitate multi-hop inference [49] or the capture of long-range dependencies [24]. Knowledge Graphs (KGs), characterized by their structured representation of interconnected entities and relationships, present a compelling alternative. Thus, research efforts are increasingly focused on developing KG-RAG systems, which aims to improve RAG performance using graph-based text representation [10, 5, 27].

Usually, a KG-RAG system operates by initially transforming source documents into graph-based representations[2]. This often involves leveraging LLMs to extract entities and relationships, which are then represented as nodes and edges within the graph. Information retrieval is subsequently performed by traversing this graph to identify nodes relevant to the query. To achieve this, traditional graph search algorithms, like PageRank [14, 9], and deep learning approaches, like GNN [53], could be employed for traversal and answer extraction [31, 8, 26, 47], these methods often fail to capture

---

[*]Corresponding author. Work done during his PhD program at Sichuan University.
[◇]Both authors contributed equally to this study.
[2]Unlike previous work leveraging established KGs [6, 42, 30, 22], this study explores KG-RAG systems that generate their knowledge graph directly from the source text, like GraphRAG [9] and LightRAG [11].

the nuanced semantic relationships embedded within the graph, leading to unsatisfactory system effectiveness [13, 10, 18]. By contrast, the LLM-guided knowledge graph traversal approaches demonstrate notable advantages [6, 42, 41], but these methods lead to substantially increased LLM invocation costs and a significant rise in irrelevant noise information. Therefore, **the effective synergy between system effectiveness and cost efficiency presents a major challenge for the practical deployment and scalability of KG-RAG systems**.

To this end, we propose REMINDRAG—**Retrieve and Memorize**, a novel LLM-guided KG-RAG system with memorization. As depicted in Figure 1, REMINDRAG first constructs a KG from unstructured texts and subsequently employs two key innovative modules: (1) *LLM-Guided Knowledge Graph Traversal with Node Exploration and Exploitation*. Beyond logical reasoning and semantic understanding capabilities of the LLM, REMINDRAG expands traversal paths using node exploration and node exploitation. This approach enables the LLM to progressively perform single-node expansion and traversal to approximate answers, achieving high-precision retrieval with more hops without requiring large-scale searches like beam search-based [54] methods. (2) *Memory Replay for Efficient Graph Traversal*. It memorizes traversal experience within KG edge embeddings, mirroring the way LLMs "memorize" world knowledge within their parameters, but in a train-free manner. It efficiently stores and recalls experience, eliminating the need to consult the LLM, thus substantially improving efficiency while preserving system effectiveness. We theoretically show that for semantically similar queries, this memorization enables edge embeddings to store sufficient traversal experience regarding these queries, facilitating effective memory replay.

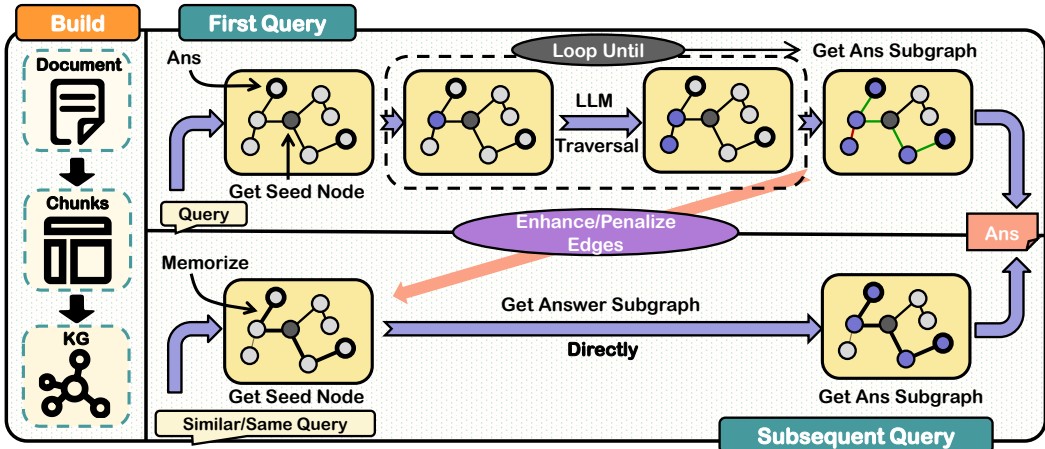

Figure 1: Overall Workflow. REMINDRAG constructs a KG from unstructured text. It memorizes LLM-guided traversal path, enabling fast retrieval when encountering similar queries subsequently.

To evaluate REMINDRAG, we conduct extensive experiments across various benchmark datasets and LLM backbones. The experimental results demonstrate that REMINDRAG exhibits a clear advantage over competing baseline approaches, achieving performance gains of 5% to 10% while simultaneously reducing the average cost per query by approximately 50%. Our analysis reveals that the key strengths of REMINDRAG lie in its inherent self-correction capability and robust memory retention. It can leverage LLMs to self-correct erroneous search paths while maintaining stable memory capabilities even when handling large-scale updates. To sum up, our work makes the following core contributions.

- We call attention to the challenge of achieving effective synergy between system effectiveness and cost efficiency, hindering the practical utility of KG-RAG systems.

- We introduce REMINDRAG, which employs LLM-guided KG traversal featuring node exploration, node exploitation, and, most notably, memory replay, to improve both system effectiveness and cost efficiency.

- We theoretically and experimentally confirm the effectiveness of our graph traversal with memory replay, demonstrating REMINDRAG's superiority over existing baselines across various benchmark datasets and LLM backbones.

## 2    Related Works

Leveraging explicit semantic relations among entities via graph-based text representation, KG-RAG systems often improve overall RAG performance [11, 9, 10, 15, 48, 12]. In these systems, information retrieval is typically performed by traversing the graph to identify nodes relevant to the query. While traditional graph search algorithms and deep learning approaches employ methods such as Personalized PageRank [14], Random Walk with Restart [45], and Graph Neural Networks (GNNs) [53, 16] for traversal and answer extraction [31, 8, 26, 47], these methods often struggle to capture the nuanced semantic relationships embedded within the graph [13, 10, 18], leading to unsatisfactory system effectiveness. In contrast, LLM-guided graph traversal methods, which prompt the LLM to make decisions about which node to visit next by virtue of their deep parsing of semantic information and flexible decision-making capabilities in the graph [43], have demonstrated effective performance [6, 42, 41, 25]. However, this approach necessitates repeated calls to the LLM, significantly increasing prompt token consumption and inference time, thus hindering practical deployment [31, 4, 15]. Moreover, recent benchmarks also indicate that their performance remains limited in complex semantic scenarios [24]. Therefore, the trade-off between system effectiveness and cost efficiency presents a major challenge for the practical deployment and scalability of KG-RAG.

To this end, we propose a novel approach that leverages LLM-guided graph traversal with node exploration, node exploitation, and, most notably, memory replay, to improve both the effectiveness and cost efficiency of KG-RAG. Furthermore, by introducing a training-free memorization mechanism, we achieve a significant advancement in graph traversal. In contrast, existing query retrieval methods are confined to traditional graph pruning [10, 5, 15] or path planning [4, 41, 11, 13, 18], with the former risking information loss and the latter offering limited efficiency gains.

## 3    Method

### 3.1    LLM-Guided Knowledge Graph Traversal with Node Exploration and Exploitation

**KG Construction**.    Building on prior works [11, 13], we begin by partitioning unstructured text into discrete chunks. Subsequently, we extract named entity triples from these chunks and construct a heterogeneous knowledge graph. This graph comprises two node types: entities and chunks. It also contains two edge types: *entity-chunk edges* (i.e., entity-to-its-source-chunk) and *entity-entity edges* (extracted triples). For specific details on knowledge graph construction, please refer to Appendix A.

**Graph Traversal**. As illustrated in Figure 2, RE-MINDRAG first identifies a set of entity nodes in the knowledge graph that exhibit the highest semantic relevance to the query based on embedding similarity, marking them as *seed nodes*. These seed nodes further form the initial subgraph $S$, which begins with no edges between them. To facilitate effective graph traversal that balances both exploration and exploitation, REMINDRAG iteratively executes two key operations for traversal path expansion: Node Exploration and Node Exploitation. The iteration continues until the LLM determines that subgraph $S$ contains sufficient information to address the query. Refer to Algorithm 1 and Appendix B for details.

---

**Algorithm 1** LLM-Guided KG Traversal

1: **Input**: Query $q$, KG $G$, Seed Count $k$, LLM

2: *// Subgraph Initialization*
3: $S \leftarrow$ top-$k$ nodes in $G$ by similarity to $q$
4: *// Path Expansion Loop*
5: **repeat**
6:     *// Node Exploration*
7:     $a \leftarrow$ LLM selects most answer-relevant node $a \in S$
8:     *// Node Exploitation*
9:     $S_a \leftarrow \{p \mid edge(a, p) \in G, p \notin S\}$
10:     $p \leftarrow$ LLM chooses optimal expansion node $p \in S_a$
11:     $S \leftarrow S \cup \{p\}$
12:     $S \leftarrow S \cup \{(a, p)\}$
13: **until** LLM confirms answer is in $S$
14: **Output**: $S$

---

- **Node Exploration Strategy**. This strategy prioritizes exploring potential nodes likely to lead to the answer, ensuring the graph traversal process is not confined to deep exploration of a single path. Instead, it achieves globally optimal search over the entire KG at low computational cost, thereby increasing the likelihood of obtaining the global optimal solution. To achieve this, during each

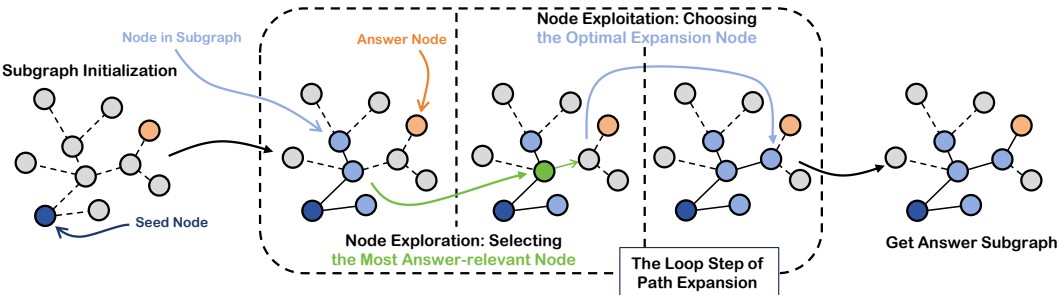

Figure 2: Illustration of LLM-Guided KG traversal in REMINDRAG. It iteratively expands the subgraph towards answer nodes through a process that combines node exploration and exploitation.

iteration, the LLM evaluates all nodes in subgraph $S$ and selects the target node $a \in S$ deemed most likely to lead to the answer.

- **Node Exploitation**. This focuses on exploiting previously explored nodes and expanding the path along those nodes to approach the answer. Specifically, given the previously selected node $a \in S$, the LLM selects the optimal expansion node $p$ from the adjacent node set $S_a$ of node $a$, considering $p$ to be the most likely to lead to the answer. At the end of the current iteration, the expansion node $p$ and its connecting path to $a$ are added to subgraph $S$ for subsequent traversal. This leverages the LLM's semantic understanding and reasoning capabilities for effective graph traversal.

### 3.2 Memory Replay for Efficient Graph Traversal

Following previous works regarding memory replay [29, 46], which refers to the recall of valuable past memories to facilitate current tasks, we design memory replay for efficient graph traversal. However, our approach diverges from prior work that typically involves recalling explicit samples. Instead, our 'memory' is embedded as weights within the KG, and it is these weights that are exclusively utilized to guide LLM inference and graph traversal. In particular, we design REMINDRAG to memorize LLM's traversal experience within the KG. Consequently, when processing identical, similar and subtly different queries, this memorized experience can be recalled before the LLM-Guided KG Traversal process, reducing the number of LLM calls and significantly enhancing the efficiency while maintaining system effectiveness.

#### 3.2.1 Memorizing the Traversal Experience within the KG

**What to Memorize**. In our work, both effective traversal paths, which lead to the correct answer, and ineffective paths are valuable, as they can provide positive and negative reinforcement, respectively, to guide the LLM's future traversal decisions. Given that the final subgraph $S$ contains a mix of both, we employ a filtering operation: For effective paths, we first use the LLM to identify the edges and chunks that contributed to generating the answer[3], and then use a Depth-First Search (DFS) [44] algorithm to extract these paths. The remaining paths are then classified as ineffective.

$$
\begin{aligned}
\textit{Weight Function}&: \delta(\boldsymbol{x}) = \frac{2}{\pi} \cos\left(\frac{\pi}{2}\|\boldsymbol{x}\|_2\right) \\
\textit{Enhance Effective Path}&: \hat{\boldsymbol{v}} = \boldsymbol{v} + \delta(\boldsymbol{v}) \cdot \frac{\boldsymbol{q}}{\|\boldsymbol{q}\|_2} \\
\textit{Penalize Ineffective Path}&: \hat{\boldsymbol{v}} = \boldsymbol{v} - \delta\left(\frac{\boldsymbol{v} \cdot \boldsymbol{q}}{\|\boldsymbol{q}\|_2}\right) \cdot \frac{\boldsymbol{v} \cdot \boldsymbol{q}}{\|\boldsymbol{q}\|_2}
\end{aligned} \tag{1}
$$

**How to Memorize**. We propose a train-free memorization method to further improve efficiency. This is achieved by memorizing traversal experiences within the KG, which are formalized as edge embeddings and updated using closed-form equations. In particular, after the LLM produces an

---
[3]We do not provide the ground truth answer to the LLM during this process. The answer here is the one produced by the LLM itself.

answer for a query, we update the edge embedding for each edge in the final subgraph $S^4$. Formally, let $v \in R^K$ denote the edge embedding, initialized as a zero vector, where $K$ is the vector dimension. For edges belonging to effective paths within $S$, we update $v$ in the direction of the query embedding $q$ generated by the LLM. Conversely, for edges belonging to ineffective paths, we reduce the component of $v$ along the query embedding $q$ (As shown in Figure 3). Equation (1) details the update rules, where $\hat{v}$ represents the updated edge embedding and $\delta$ is the update weight function, which is essentially a cosine function based on the vector's norm.

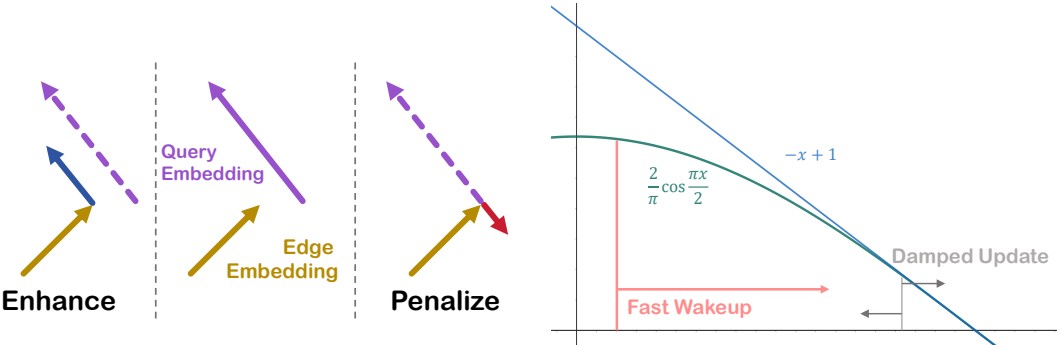

Figure 3: Schematic Diagram of Memorizing.

Figure 4: Understanding Fast Wakeup & Damped Update via weight function $\delta$.

**Intuitions Behind Memorizing Process**. Our closed-form memorization rules, as outlined in Equation (1), exhibit two key characteristics: rapid learning of traversal experiences and sustained stability across multiple memorization events. These characteristics are achieved through the *Fast Wakeup* and *Damped Update*, respectively.

- Fast Wakeup. When the norm of the edge embedding $v$ is small (typically in the initial all-zero state), it indicates that $v$ retains little memorized information. In this state, we aim for the edge embedding to rapidly learn the traversal experience with only a few updates. As shown in Figure 4, when the norm is small, the $\delta$ function induces a substantial directional update to the edge embedding, thus enabling Fast Wakeup.

- Damped Update. Once the edge embedding $v$ has learned substantial information (i.e., the norm of $v$ is large), our objective is to maintain the stability of the edge embedding, preventing it from easily losing its learned information due to subsequent updates. As illustrated in Figure 4, the $\delta$ function induces only minor updates when the norm is large, exhibiting resistance to change while retaining the ability to correct errors through gradual forgetting.

### 3.2.2 Efficient Subgraph Expansion via Memory Replay

To enhance traversal efficiency, the memory replay mechanism enriches the initial subgraph $S$ prior to the LLM-guided path expansion (detailed in Algorithm 1). This mechanism achieves rapid subgraph expansion by incorporating potentially relevant nodes and edges from historical traversal experiences, thereby reducing reliance on iterative LLM calls.

Formally, let $q$ represent a user query and $G$ denote a knowledge graph, where each edge $E_{ab}$ connecting nodes $a$ and $b$ possesses an associated, updatable embedding vector $v_{ab}$. We define $\text{emb}(\cdot)$ as the text embedding function and $\text{sim}(\cdot, \cdot)$ as the node embedding similarity function. The Subgraph Initialization, detailed in Algorithm 1, begins by generating a graph comprising only seed nodes. From each seed node, a recursive Depth-First Search (DFS) is performed to expand the subgraph by adding traversed nodes and edges. This DFS operation proceeds as follows: For a given current node $N_{\text{from}}$ and its unvisited neighbor $N_{\text{to}}$, the system assesses $N_{\text{to}}$'s relevance to both $N_{\text{from}}$ and the query $q$, estimating its potential contribution to the final answer. If this relevance surpasses a predefined threshold $\lambda$, $N_{\text{to}}$ and its connecting edge are incorporated into subgraph $S$, and the DFS process is recursively invoked on $N_{\text{to}}$. The relevance metric $w_{N_{\text{from}}, N_{\text{to}}}$, as defined in this work, consists of two

---

[4]Each edge in KG has an associated embedding vector, which is subject to updating if that edge is included in the final subgraph.

components: (1) the semantic relevance between nodes $N_{\text{from}}$ and $N_{\text{to}}$[5]; and (2) the alignment of traversing the edge with satisfying query $q$, which constitutes REMINDRAG's core memory function. The hyperparameter $\alpha$ adjusts the relative importance of these two terms.

$$w_{N_{from},N_{to}} = \alpha \cdot \text{sim}\big(\text{emb}(N_{from}), \text{emb}(N_{to})\big) + (1-\alpha) \cdot \frac{\text{emb}(q) \cdot v_{N_{from},N_{to}}}{\|\text{emb}(q)\|} \qquad (2)$$

After subgraph expansion via memory replay, the subgraph is expected to contain sufficient information for answering the input query. However, if the LLM judges that the expanded subgraph still lacks sufficient information, REMINDRAG continues the Path Expansion Loop, a process that can be regarded as verifying and correcting memorized information.

## 4 Theoretical Analysis

Our theoretical analysis (cf. **Appendix F** for a detailed proof) of the memory capacity demonstrates that, given a set of queries exhibiting some degree of semantic similarity, our memorization rules in Equation (1) enable edge embeddings to memorize these queries and store sufficient information, thereby enabling effective memory replay. Formally, when the embedding dimension $d$ is sufficiently large, given the threshold $\lambda$ in Section 3.2.2, for any given edge, when the maximum angle $\theta$ between all query embedding pairs within a query set satisfies Equation (3), the edge can effectively memorize the semantic information within that query set.

$$\theta \leq \lim_{d \to \infty} \left[ 2 \arcsin \left( \sqrt{\frac{1}{2}} \sin(\arccos(\lambda)) \right) \right] \qquad (3)$$

**Remark 1** *To satisfy the storage requirements*[6]*, the theoretical upper bound for $\lambda$ is determined to be 0.775. This determination aligns with existing research [37], which establishes a cosine similarity threshold of 0.6 as the criterion for assessing semantic similarity.*

**Remark 2** *Our assumption that the embedding dimension $d$ is infinitely large is based on the fact that linear embedding models do have very high dimensions. For the function $\sqrt{\frac{d+1}{2d}}$, when $d$ is greater than around 100 (currently, the embeddings of an LLM often exceed 100 dimensions), it can actually be approximated to $\frac{1}{2}$. This approximation thus holds significant practical utility. See our empirical experiments in Section 5.3.2.*

## 5 Experimental Analysis

### 5.1 Experiment Setup

We conduct extensive experiments across various benchmark datasets and LLM backbones. For *implementation details* and *additional analysis*, refer to Appendix D and Appendix E.1, respectively.

**Tasks & Datasets.** To comprehensively evaluate the performance of the proposed REMINDRAG, we follow previous works [13, 24] and conduct experiments on three benchmark tasks: Long Dependency QA, Multi-Hop QA, and Simple QA. This enables systematic assessment of retrieval effectiveness and efficiency across various scenarios. Refer to Appendix C for details.

1) Long Dependency QA: We resort to the LooGLE dataset [24]. It requires the system to identify implicit relationships between two distant text segments, testing the RAG system's long-range semantic retrieval capability. The long-dependency QA task in the LooGLE dataset (with evidence fragments spaced at least 5,000 characters apart) effectively evaluates the long-range dependency capture performance of RAG solutions in single-document scenarios.

---

[5]This component enables user control over REMINDRAG's memory sensitivity.

[6]While this provides a solution under theoretical conditions for parameter configuration, practical applications require consideration of multiple factors when setting parameters. Additional details are in Appendix D.4.

2) Multi-Hop QA: We resort to the HotpotQA dataset [49]. It focuses on complex question-answering scenarios requiring multi-step reasoning, demanding the model to chain scattered information through logical inference.

3) Simple QA: This task leans towards traditional retrieval tasks, emphasizing the model's ability to extract directly associated information from local contexts. We adopt the Short Dependency QA from the LooGLE [24] dataset as a representative example.

**Evaluation Metrics**. (1) Effectiveness evaluation. Following the evaluation protocol of recent benchmark [24], for each question, we employ LLM-as-judgment (GPT-4o) [52] to determine the answer accuracy: whether the model-generated answer adequately covers the core semantic content of the ground truth through semantic coverage analysis. Refer to Appendix D.6 for more details. (2) Cost-efficiency evaluation. We use the average number of tokens consumed by the LLM per query during traversal as our cost-efficiency metric.

**Baselines.** We consider following baselines[7]: (1) Traditional retrieval methods: BM25 [38] and NaiveRAG [23]. (2) KG-RAG systems using graph search algorithms: GraphRAG [9], LightRAG [11] and HippoRAG2 [13]. (3) LLM-guided KG-RAG system: Plan-on-Graph [6].

**Backbone.** We employed two backbones: (1) GPT-4o-mini [34], a highly cost-effective and currently the most popular backbone. (2) Deepseek-V3 [28], the latest powerful open-source dialogue model.

## 5.2 Understanding the Effectiveness and Cost-Efficiency of REMINDRAG

### 5.2.1 System Effectiveness Evaluation

As shown in Table 1, REMINDRAG consistently outperforms the baselines across different LLM backbones, with performance fluctuations confined to a small range. Particularly noteworthy is the significant performance improvement achieved on Long Dependency QA and Multi-Hop QA tasks (+12.08% and +10.31% over the best baseline on average), while simultaneously maintaining high accuracy on Simple QA tasks (+4.66% over the best baseline on average). This phenomenon indicates that REMINDRAG not only exhibits strong fundamental retrieval capabilities but also effectively addresses long-range information retrieval and multi-hop reasoning challenges. This advantage primarily stems from two factors: (1) By leveraging the reasoning capabilities of LLMs, the system can perform semantic-driven active reasoning on KGs, enabling more precise answer retrieval. (2) The balanced node exploration and exploitation strategy employed during LLM-guided KG traversal allows for the achievement of both globally and locally optimal search results.

| QA Type | Long Dependency QA | | Multi-Hop QA | | Simple QA | |
|---|---|---|---|---|---|---|
| Backbone | GPT-4o-mini | Deepseek-V3 | GPT-4o-mini | Deepseek-V3 | GPT-4o-mini | Deepseek-V3 |
| **REMINDRAG** (*Ours*) | **57.04%** | **59.73%** | **74.22%** | **79.38%** | **76.67%** | **77.01%** |
| Plan-on-Graph [6] | 27.78% | 19.46% | 58.51% | 47.87% | 38.26% | 32.89% |
| HippoRAG2 [13] | 39.60% | 38.26% | 68.04% | 64.95% | 73.08% | 71.28% |
| LightRAG [11] | 37.58% | 53.02% | 45.36% | 53.60% | 49.49% | 62.82% |
| GraphRAG [9] | 26.03% | 9.86% | 58.70% | 22.95% | 21.96% | 7.20% |
| NaiveRAG [23] | 36.91% | 46.98% | 58.76% | 56.70% | 47.95% | 66.41% |
| BM25 [38] | 22.82% | 33.56% | 50.52% | 54.64% | 45.13% | 52.56% |
| LLM Only (w/o RAG) | 16.11% | 29.53% | 40.21% | 54.64% | 18.21% | 30.00% |

Table 1: Effectiveness Performance (i.e., Answer Accuracy).

### 5.2.2 Evaluation on System Cost-Efficiency and Memorization

**Setups**. To evaluate the cost efficiency (along with its effectiveness) of REMINDRAG, we conducted multiple sets of comparative experiments under identical conditions. Here, given a dataset $A$ containing documents and user queries, we consider three setups: (1) Same Query: The model initially has already been evaluated on dataset $A$ and has memorized information from $A$ and is

---

[7]For additional analysis on baselines, please refer to Appendix E.2.

subsequently re-evaluated again on the same dataset. (2) Similar Query: The model is re-evaluated again on a dataset $A'$ whose queries are semantically equivalent paraphrases of those in $A$ (cf. Appendix C.3 for implementation). (3) Different Query: The model is re-evaluated again on a dataset $A''$ whose queries are distinct from those in $A$ but share similar questions (cf. Appendix C.4 for implementation). Importantly, of all the datasets we tested, *only the Multi-Hop QA dataset satisfies the criteria for the Different Query setup*[8]. Additionally, to evaluate the memorization capability of REMINDRAG comprehensively, we further consider *multi-turn memorization*. This involves initially evaluating REMINDRAG on the same dataset $A$ multiple times (i.e., memorizing $A$ multiple times).

As shown in Table 2, REMINDRAG demonstrates significant cost reduction when utilizing memory replay. Furthermore, after the multiple-turn memorization, REMINDRAG achieves a 58.8% reduction in token consumption compared to the initial trial, on average. These optimizations are primarily due to REMINDRAG's ability to learn and memorize traversal experiences via dense embedding techniques. When encountering identical or semantically similar queries, REMINDRAG can rapidly identify relevant structures based on its memories to initialize the subgraph $S$, thereby reducing the frequency of LLM calls and significantly lowering costs. Additionally, we observe that when handling multiple distinct questions pertaining to the same underlying document (i.e., the Different Query scenario), REMINDRAG effectively stores and utilizes traversal knowledge acquired from various queries. This observation aligns with our theoretical findings presented in Section 4, which state that given a set of queries exhibiting semantic similarity, our memorization rules enable edge embeddings to effectively memorize these queries and store sufficient information, thereby enabling effective memory replay.

| QA Type | Long Dependency QA | | | | Multi-Hop QA | | | | | |
| | Same Query | | Similar Query | | Same Query | | Similar Query | | Different Query | |
| Methods | Accuracy | Tokens | Accuracy | Tokens | Accuracy | Tokens | Accuracy | Tokens | Accuracy | Tokens |
|---|---|---|---|---|---|---|---|---|---|---|
| REMINDRAG | | | | | | | | | | |
| *-3-turn Memorization* | 60.31% | 6.71K | 57.25% | 7.02K | 87.62% | 5.89K | 78.72% | 6.02K | 77.66% | 9.76K |
| *-2-turn Memorization* | 58.01% | 7.55K | 54.96% | 9.85K | 84.04% | 6.56K | 73.40% | 9.34K | 74.47% | 9.50K |
| *-1-turn Memorization* | 56.48% | 9.68K | 55.03% | 13.98K | 79.78% | 7.73K | 75.53% | 9.88K | 72.34% | 10.57K |
| *-No Memorization* | 57.04% | 14.91K | / | / | 74.22% | 10.16K | / | / | / | / |
| Plan-on-Graph | 27.78% | 30.29K | 30.87% | 39.08K | 58.51% | 5.53K | 61.19% | 7.01K | 62.67% | 5.45K |
| HippoRAG2 | 39.60% | / | 37.58% | / | 68.04% | / | 63.91% | / | 69.44% | / |

Table 2: Evaluation on System Cost-Efficiency and Memorization under three query types and multi-turn memorization: GPT-4o-mini is used as the backbone. "Tokens" here refer to the average number of tokens consumed by the LLM per query during traversal.

## 5.3 Understanding the Characteristics of REMINDRAG

This section explores features of REMINDRAG, including its self-correction capabilities and memory stability. For more in-depth analyses concerning graph construction efficiency, the effect of contextual data, and the influence of varying max hop counts, please refer to Appendix E.1.

### 5.3.1 Self-Correction of REMINDRAG

As shown in Table 2, REMINDRAG achieves a 5%-10% performance improvement after multi-turn memorization, revealing the potential for self-correction in graph traversal. For a better understanding of this behavior, we provide a case study[9] presented in Figure 5. The case study demonstrates that when REMINDRAG initially provides an incorrect answer subgraph, the system can still penalize irrelevant nodes in the knowledge graph based on the memorization rules (Equation (3)) without explicitly receiving the correct answer. In subsequent queries, REMINDRAG switches to traversing the memory-optimized subgraph, thereby achieving self-correction of errors. Furthermore, Figure 6 showcases REMINDRAG's robust self-correction in the "Different Query" setting. When handling queries that are embedding-similar but fundamentally distinct, the candidate answer subgraphs retrieved via similarity matching fail to address the current query. In such cases, REMINDRAG fully leverages the deep semantic understanding capabilities of LLMs to identify mismatches between the

---

[8]We use the "hotpot_dev_distractor" version, which contains intentionally designed distractors.

[9]Demonstration screenshots from our operational web interface are available in Appendix G.2.

candidate answer subgraphs and the given query, and then autonomously triggers the LLM-based graph traversal mechanism to re-execute the reasoning process. REMINDRAG still demonstrates excellent performance and remarkable cost reduction on these challenging problems.

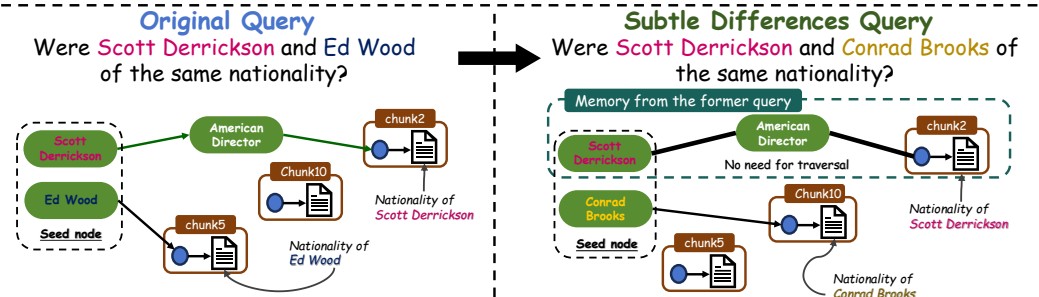

Figure 5: The case of memory replay under the Same Query setting. In the first search, no relevant info about 'Ed Wood' is found, but instead some unrelated details about 'Scott Derrickson' appear. After the search, the system enhances useful paths and penalizes irrelevant ones. The second search then focuses on the subgraph with the correct info about 'Scott Derrickson', achieving self-correction.

### 5.3.2 Memory Stability

This section aims to evaluate the memory stability of REMINDRAG in more complex scenarios.

**Impact of Distinct Queries**. Unlike the multi-turn memorization task described in Section 5.2.2 (where RE-MINDRAG repeatedly memorizes the same dataset), this experiment requires REMINDRAG to alternately process and memorize heterogeneous datasets associated with distinct queries. The alternating update procedure frequently involves cyclic operations of enhancing and penalizing specific edges, rigorously val-

| REMINDRAG w/ multi-turn memo. | QA Type | Accuracy | Tokens |
|---|---|---|---|
| *-5-turn Memorization* | Different | 77.66% | 6.33K |
| *-4-turn Memorization* | Origin | 80.85% | 5.72K |
| *-3-turn Memorization* | Different | 76.60% | 6.75K |
| *-2-turn Memorization* | Origin | 78.72% | 6.91K |
| *-1-turn Memorization* | Different | 72.34% | 10.57K |
| *-No Memorization* | Origin | 74.22% | 10.16K |

Table 3: Accuracy of REMINDRAG under complex multi-turn memorization settings (Multi-Hop QA & GPT-4o-mini). "Origin" represents the original question, while "Different" denotes the modified question (as shown in **Setup** (3) in Section 5.2.2).

idating REMINDRAG's memory stability under extreme conditions. As shown in Table 3, RE-MINDRAG demonstrates sustained promising performance and cost reduction, which is a direct result of our designed "Fast Wakeup" and "Damped Update" mechanisms detailed in Section 3.2. During multi-round memorization, REMINDRAG demonstrates rapid content memorization while preserving stability in retaining effective memories, highlighting its potential for handling dynamic update patterns in complex scenarios.

Figure 6: The case of memory replay under the Different Query setting. In this case, a person's name is replaced in the second query, resulting in different answer subgraphs. However, the LLM is able to detect the incompleteness of the candidate answer subgraph and restart the traversal process. A complete demonstration of the entire process can be found in Appendix G.1.

| Dimension | 768 | 384 | 192 | 96 | 48 | 24 | 12 |
|---|---|---|---|---|---|---|---|
| *-No Memorization* | 57.04% | 56.37% | 55.36% | 54.69% | 54.36% | 52.78% | 52.08% |
| *-1-turn Memorization* | 56.48% | 52.34% | 54.03% | 53.02% | 52.97% | 52.95% | 52.55% |
| *-2-turn Memorization* | 58.01% | 57.71% | 55.03% | 53.35% | 53.28% | 53.42% | 53.15% |

Table 4: Dimensional Truncation Experiment. The horizontal axis represents the dimension truncation ratio. The vertical axis denotes the number of updates. The values in the table correspond to the answer accuracy, and all experiments were conducted with the same default parameters.

**Impact of Embedding Dimensions on Memory Capacity**. While our theoretical analysis in Section 4 establishes REMINDRAG's memory capacity, it hinges on an approximation where the LLM's embedding dimension approaches infinity (details in Appendix F). Although Appendix F argues this approximation remains practically useful for large dimensions (e.g., 100), this section empirically demonstrates the robustness of our memory capacity even with varied, truncated dimensions. As detailed in Table 4, experiments with reduced embedding dimensions show negligible impact on the overall system. We hypothesize this resilience stems from the embedding model's Matryoshka Representation Learning [21], allowing normal operation even at low dimensions.

# 6   Conclusion

In this paper, we introduce REMINDRAG, a novel approach that leverages LLM-guided knowledge graph traversal featuring node exploration, node exploitation, and, most notably, memory replay, to improve both the effectiveness and cost efficiency of KG-RAG. We theoretically demonstrate that the memorization capability of REMINDRAG is strong. It could store sufficient information to enable effective memory replay. Our experimental results show that REMINDRAG achieves promising performance compared to baselines across multiple benchmark datasets and LLM backbones. In-depth analysis reveals that REMINDRAG effectively memorizes the LLM's traversal experience, reducing costs and enhancing efficiency in subsequent queries under various query settings (i.e., Same, Similar, and Different). Going forward, while the initial graph traversal of REMINDRAG still requires multiple LLM calls, we plan to initialize the model with pre-existing, well-curated FAQs specific to the deployment domain or scenario. This will provide the model with a strong baseline memory, capable of handling common situations and accelerating the graph traversal process.

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

# A Heterogeneous Knowledge Graph in REMINDRAG

## A.1 Heterogeneous Knowledge Graph Formalization

Inspired by prior works [3, 50, 11, 13], in the automated knowledge graph construction framework, our designed heterogeneous knowledge graph consists of an entity node set $\mathcal{E}$, an anchor node set $\mathcal{A}$ (storing text chunk titles), a text chunk set $\mathcal{C}$, and relational connections. The subgraph formed by entity nodes is composed of "entity-relation-entity" triples extracted by the large language model (LLM). Each anchor node $\mathcal{A}_i$ serves as a summary of the corresponding text chunk $\mathcal{C}_i$, forming a one-to-one mapping relationship: 1) $(\mathcal{A}_i, \mathcal{C}_i)$ $\forall i \in \{1, 2, \ldots, n\}$. When an entity node $\mathcal{E}_i$ appears in a certain text chunk $\mathcal{C}_j$, the corresponding

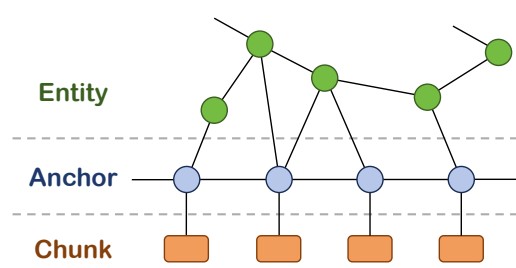

Figure 7: Illustration of Our KG.

anchor node $\mathcal{A}_j$ establishes a connection with $\mathcal{E}_i$. Meanwhile, anchor nodes construct the **Contextual Skeleton Network** through ordered chain connections, with directed edges defined as: 2) $(\mathcal{A}_i, \mathcal{A}_{i+1})$ for $i = 1, 2, 3, \ldots, n-1$ (for $n$ text chunks), forming a chain structure $\mathcal{A}_1 \leftrightarrow \mathcal{A}_2 \leftrightarrow \cdots \leftrightarrow \mathcal{A}_n$, as shown in Figure 7. This backbone network with chain connections of anchor nodes explicitly models the contextual dependencies among text chunks $\mathcal{C}$, preserving document context relationships in the graph and providing support for subsequent LLM traversals. The chunk-anchor mapping is designed to reduce computational costs during the Memorize-Recall process. This architecture allows REMINDRAG to leverage contextual linkage relationships without requiring a chunk to be included in the enhanced answer subgraph. For ablation studies on our contextual linking mechanism, see Appendix E.1.

## A.2 Automated Knowledge Graph Building

Our knowledge graph extraction pipeline follows the conventional workflow: (1) Text Chunking, (2) Named Entity Recognition, (3) Relation Triple Extraction, and (4) Knowledge Graph Building. The examples of our KG are illustrated in Figure 16.

**Text Chunking.** Our implementation supports multiple text chunking methods, including the basic token-count-based chunking, meta chunking [51], and LLM-based chunking [39]. For consistent evaluation, we uniformly employ the basic token-count-based chunking.

**Named Entity Recognition.** We use the prompt shown in Table 18 to extract entities from each text chunk.

**Relation Triple Extraction.** We apply the prompt presented in Table 19 to extract relations from text chunks after entity extraction.

**Knowledge Graph Building.** We create a chunk node and an anchor node for each text chunk. For each entity, we establish a separate node. When the embedding similarity between two entities exceeds our predefined threshold (**0.7**), we consider them synonyms and merge them. Each chunk connects to its corresponding anchor. Anchor nodes of text chunks from the same paragraph connect via the **Contextual Skeleton Network**. Entities link to the chunks where they occur. Related entities connect through relation edges.

# B LLM-Guided Knowledge Graph Traversal with Node Exploration, Node Exploitation and Memory Replay

**Overview**. Our entire **LLM-Guided Knowledge Graph Traversal** can be divided into the following steps: (1) Pre-query question processing; (2) Fast subgraph extraction using Memory Replay; (3) LLM-Guided KG traversal (optional, only if the subgraph obtained in step 2 cannot answer the question); (4) Memorizing LLM-Guided KG traversal (optional, only if step 3 was executed); (5) Generating the answer using the obtained results. As depicted in Figure 8.

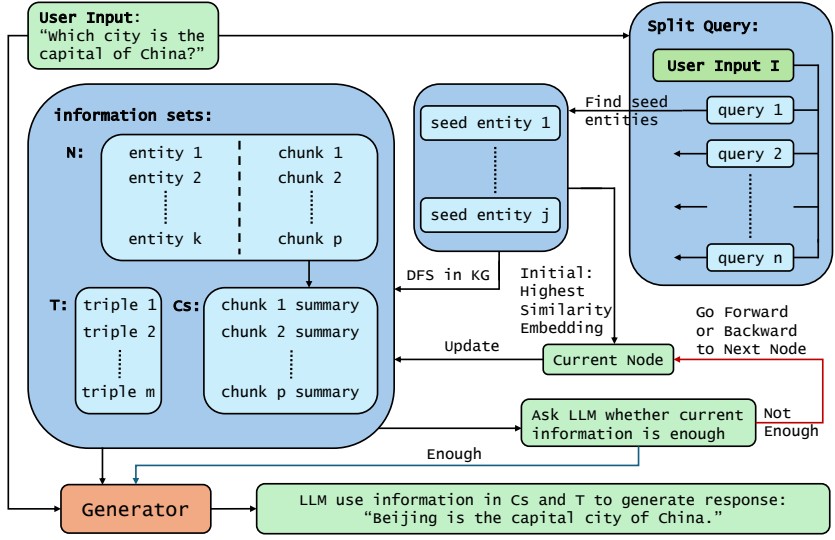

Figure 8: LLM-Based Knowledge Graph Traversal Process

### B.1 Pre-query Question Processing

When a user inputs a request $I$, our model first queries the large language model (LLM) to determine whether external knowledge in the knowledge graph (KG) is required to respond. (Determine whether it is necessary to conduct a search process). If affirmative, the model decomposes the input into several subqueries $Q = [q_1, q_2, \ldots, q_n] = \text{SplitQuery}(I)$. For consistency with baseline comparisons, the query decomposition was disabled during experiments shown in Section 5.

### B.2 Fast subgraph extraction using Memory Replay

Then every subquery is matched with the KG by embedding similarity to identify the top-$K$ seed entities. Starting from these seed entities, we perform a thresholded DFS-based subgraph expansion (as shown in Section 3.2.2) governed by edge embedding $w_{A,B}$ and a hyperparameter threshold $\lambda$. This traversal initializes three information sets:

$N = [e_1, e_2, \ldots, e_k, c_1, c_2, \ldots, c_p]$: Entities and chunks in Subgraph.
$Cs = [s_1, s_2, \ldots, s_p]$: Semantic summaries of the chunks in $N$.
$T = [t_1, t_2, \ldots, t_m]$: Triples (edges) in Subgraph.

### B.3 LLM-Guided KG Traversal

The seed entity with the highest similarity is designated as the initial $node_c$. We iteratively ask the LLM to assess whether the current information sets $Cs$, $T$ contain sufficient information. If inadequate, the LLM is prompted either as Table 12:

**Forward**: to select a new node $node_{next}$ from linked nodes of $node_c$. $node_{next}$ is added to $N$. If it is a chunk, the summary is processed and added to $Cs$. The corresponding knowledge triple connecting $node_c$ to $node_{next}$ is added to $T$.
**Backward**: to select an encountered node in $N$. No previously collected elements in $N$, $Cs$, or $T$ are removed.

This step essentially combines the "Node Exploration" and "Node Exploitation" from Section 3.1 into a single LLM call to reduce costs. When the LLM selects "forward," it essentially treats the most answer-relevant node as $node_c$ and then chooses the optimal expansion node. When the LLM selects "backward," it updates the current $node_c$ to a new most answer-relevant node.

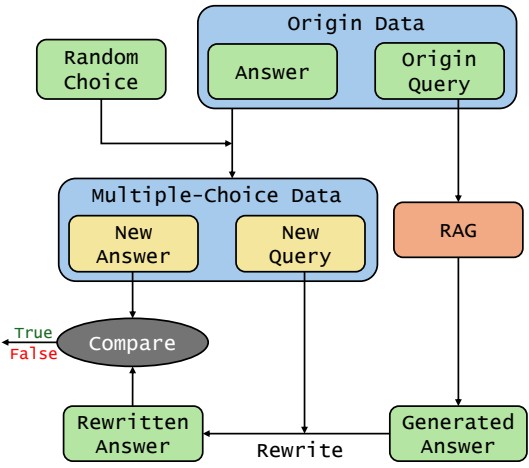

Figure 9: LooGLE Test Pipeline

## B.4 Memorizing LLM-Guided KG Traversal

After the query concludes, we use Table 13 to select the nodes or edges in the answer subgraph that contribute to the answer. We then perform a DFS search to enhance the paths leading to these nodes or edges while penalizing other paths. The detailed process is shown in Section 3.2.

## B.5 Generating the Answer Using the Obtained Results

Finally, once the information is deemed sufficient, the LLM generates a response to the user query based on the information in $Cs$ and $T$.

Our framework utilizes an ensemble of prompt instructions. Due to space constraints, we focus specifically on illustrating the fundamental 'node selection'(LLM-Guided KG Traversal) prompt (Table 12), with the complete set of supplementary prompts available in our open-source code repository.

## C  Experimental Dataset

The QA pairs in the Free-response Question format of the original LooGLE dataset suffer from substantial semantic coverage bias issues. Specifically, when a generated answer contains more comprehensive information than the reference answer, the GPT-4 discriminator in LooGLE (which relies on semantic equivalence) frequently marks the question as incorrect, because the real answers in LooGLE rarely achieve complete semantic coverage of the generated answers—even when the generated answers are substantively correct. Therefore, we have reformatted these questions into a multiple-choice pattern using LLMs: first, the original question is provided to the RAG system, and based on its returned content and the reformatted multiple-choice questions, GPT-4o is used for selection. This approach avoids information leakage while ensuring the objectivity of answer evaluation (see Appendix C.1 for the dataset reformatting methodology). Some dataset samples can be found in Appendix C.2. Meanwhile, the rewriting process for the "Similar" and "Different" fields in Section 5.2 is also demonstrated in Appendix C.3 and Appendix C.4.

## C.1  Dataset Transformation Methodology

The standardized experimental pipeline employed in this study is illustrated in Figure 9. For each question in the LooGLE dataset [24], we process the original data through the following procedure: The "question", "answer", and "evidence" fields from the raw data are provided as input to the GPT-4o [35] model, which is instructed to generate a multiple-choice question with four options. The correct option is derived from the given answer, while three plausible distractors are constructed based on the "evidence" field. The specific prompt template for question transformation can be found in Table 14.

To prevent potential information leakage and ensure fair evaluation, we implement a two-stage processing strategy: First, the original question is fed into the retrieval system; subsequently, GPT-4o selects the most appropriate answer from the generated options based on the retrieval results (with the option to abstain if confidence is insufficient). The detailed prompt template for option selection is provided in Table 17. This design maximizes the objectivity and scientific rigor of the evaluation process. Some dataset samples are illustrated in Appendix C.2.

## C.2  Multiple-Choice Format of the LooGLE Dataset

This section presents several examples from the multiple-choice version of the LooGLE dataset for reference. As shown in Figure 10a, for the original question, both REMINDRAG and other baseline methods face challenges in evaluation because all four options are presented in a (number-number) pair format. Consequently, when verifying answers, LLMs struggle to determine whether the model's response refers to an option or the actual answer.

Additionally, as illustrated in Figure 10b, we observe that nearly all methods incorporate the evidence "but is available second-hand." into their responses. However, during answer verification, the model lacks awareness of whether this information is factually correct. Since the LooGLE reference answer does not include this detail, such responses are incorrectly judged. These cases are not uncommon, which motivated our modifications to the dataset.

(a) Ambiguous Answer Reference    (b) Insufficient Answer Information

Figure 10: LooGLE Dataset Samples

## C.3  Similar Question Rewriting Details

For questions in "Long Dependency QA" and "Hotpot QA", we employed the prompt templates from Table 15 to rewrite them. While preserving the essence of each question, we aimed to maximize the surface-level differences between the rewritten and original versions. A concrete example of such rewriting is provided in Figure 11.

> **ORIGIN QUESTION:**
> Among 10 districts of Barcelona, Which district has the highest population density?
>
> **SIMILAR QUESTION:**
> Which district in Barcelona has the greatest population density among its 10 districts?

Figure 11: An Example of Similar Question in LooGLE

> **ORIGIN QUESTION:**
> What science fantasy young adult series, told in first person, has a set of companion books narrating the stories of enslaved worlds and alien species?
>
> **SUBTLE DIFFERENCE QUESTION:**
> Which young adult science fantasy series, narrated from multiple perspectives, includes companion books detailing the histories of enslaved worlds and alien species?

Figure 12: An Example of Subtle Difference Question in Hotpot QA

## C.4  Subtle Differences Question Rewriting Details

For questions in "Hotpot QA", we used the prompt templates from Table 16 to rewrite them. While maintaining as much semantic similarity as possible, we altered the questions so that their answers differed from the original versions. A specific rewriting example is shown in Figure 12.

## D  Implementation Details

### D.1  Experimental Configuration

To ensure the reproducibility of our experiments, all tests were conducted with consistent parameter configurations. The random seed for the large language model was fixed to **123**, and the temperature parameter was set to **0** to eliminate randomness in the generation process. Additionally, GPT-4o is employed as our LLM-based evaluator. Furthermore, all dense embedding computations employed the "**nomic-ai/nomic-embed-text-v2-moe**" model [33] as the foundational embedding model. For tokenization operations, we uniformly adopted the "**nomic-ai/nomic-embed-text-v2-moe**" as tokenizer, with all token-based chunks standardized to **750** tokens in length. For detailed parameter settings of other experiments, please refer to Appendix D.4.

### D.2  Compute Resources

In our experiments, we employed **GPT-4o-mini** [34] and **Deepseek-V3** [28] as the underlying large language models (LLMs). (All operations utilizing large language models (GPT-4o, GPT-4o-mini, Deepseek-V3) are completed by invoking APIs). Notably, the proposed experiments can be readily replicated on ordinary machines, ensuring broad accessibility and reproducibility for the research community. Experiments were executed on a dedicated research workstation configured with an AMD Ryzen 7 7800X3D 8-Core Processor, an NVIDIA GeForce RTX 4070 Ti SUPER GPU, and 64 GB of DDR5 RAM.

### D.3  Evaluation Metrics

Our efficiency is quantified in our paper through token consumption. Our decision not to use and can not use precise wall-clock time latency as a direct metric stems from two key reasons: (1) the strong correlation between LLM inference time and the number of output tokens, and (2) the inherent uncertainties and fluctuations caused by network latency during API calls to LLMs, which would introduce inaccuracies into direct time measurements.

### D.4 Parameter Configuration

The system involves the following key parameters:

- **Node Correlation Weight** ($\alpha$): This parameter adjusts the system's reliance on edge embedding for determining strong links. Experimental validation suggests setting $\alpha$ within the range of 0.1–0.2. In this study, we adopt $\alpha = \mathbf{0.1}$.
- **Strong Connection Threshold** ($\lambda$): As discussed in Section 4, we typically set it to a value below 0.775, but this is only under theoretical conditions. In practice, various factors should be comprehensively considered—setting it too low may lead to increased retrieval costs due to a lower memory threshold outweighing the noise reduction, while setting it too high may result in reduced memory capacity. Through empirical testing, we consider values between 0.5 and 0.75 to be reasonable. In the experiments of this paper, we select this value as **0.55**.

Other critical parameters include:

- **Synonym Similarity Threshold**: Used during knowledge graph construction to merge entities whose embedding similarity exceeds this threshold. This value should be adjusted according to the specific embedding model characteristics. Our experiments employ **0.7** as the default value.
- **Maximum Hop Count**: Controls the maximum number of nodes (hops) during subgraph expansion for word queries. We set this parameter to **10** to balance computational efficiency and information completeness. For other baselines with similar configurations, we uniformly apply the same setting for fair comparison.
- **Question Decomposition Limit**: Determines the maximum number of sub-questions for semantic decomposition of user queries. To maintain comparability with baseline methods, we set this to **1** (i.e., preserving the original question without decomposition).
- **Initial Seed Node Count**: Specifies the number of seed nodes in the query initialization phase. We configure this as **2** seed nodes (selecting the two most query-relevant nodes), consistent with relevant baseline studies.

### D.5 Implementation of Baselines

**Traditional retrieval methods:**

- **BM25** [38]. A classic information retrieval algorithm improving on TF-IDF [40], it estimates document-query relevance via a probabilistic framework. It effectively handles long documents and short queries by integrating term frequency (TF), inverse document frequency (IDF), and document length normalization to calculate relevance scores, enhancing retrieval accuracy through adaptive term weighting. Following its principles, we have realized the algorithm.
- **NaiveRAG** [23]. The NaiveRAG paradigm, an early "Retrieve-Read" framework, involves indexing multi-format data into vector-encoded chunks stored in a database, retrieving top-K similar chunks via query vector similarity, and generating responses by prompting large language models with queries and selected chunks. We have developed the corresponding code based on its principles.

**Traditional KG-RAG systems**:

- **HippoRAG2** [13]. This framework enhances retrieval-augmented generation (RAG) to mimic human long-term memory, addressing factual memory, sense-making, and associativity. It uses offline indexing to build a knowledge graph (KG) with triples extracted by an LLM, integrating phrase nodes (concepts) and passage nodes (contexts) via synonym detection. Online retrieval links queries to KG triples using embeddings, filters irrelevant triples with an LLM (recognition memory), and applies Personalized PageRank (PPR) on the KG for context-aware passage retrieval. Improvements include dense-sparse integration, deeper query-to-triple contextualization, and effective triple filtering.
- **LightRAG** [11]. This method addresses limitations of existing RAG systems (e.g., flat data representations, insufficient contextual awareness) by integrating graph structures into text indexing and retrieval. It employs a dual-level retrieval paradigm: low-level retrieval for specific entities/relationships and high-level retrieval for broader themes, combining graph structures with vector

representations to enable efficient keyword matching and capture high-order relatedness. The framework first segments documents, uses LLMs to extract entities/relationships, and constructs a knowledge graph via LLM profiling and deduplication for efficient indexing.

- **GraphRAG** [9]. This approach aims to enable large language models (LLMs) to perform query-focused summarization (QFS) over large private text corpora, especially for global sensemaking queries that require understanding of the entire dataset. It constructs a knowledge graph in two stages: first, an LLM extracts entities, relationships, and claims from source documents to form a graph where nodes represent key entities and edges represent their relationships. Second, community detection algorithms partition the graph into a hierarchical structure of closely related entity communities. The LLM then generates bottom-up community summaries, with higher-level summaries recursively incorporating lower-level ones. Given a query, GraphRAG uses a map-reduce process: community summaries generate partial answers in parallel (map step), which are then combined and summarized into a final global answer (reduce step).

For these three baselines, we use their default hyperparameters (except for chunk size, which is uniformly set to 750, consistent with REMINDRAG) and prompts.

**LLM-guided KG-RAG system**:

- **Plan-on-Graph (PoG)** [6]. This paradigm aims to address the limitations of existing KG-augmented LLMs, such as fixed exploration breadth, irreversible paths, and forgotten conditions. To achieve this, it decomposes the question into sub-objectives and repeats the process of adaptive reasoning path exploration, memory updating, and reflection on self-correcting erroneous paths until the answer is obtained. PoG designs three key mechanisms: 1) Guidance: Decomposes questions into sub-objectives with conditions to guide flexible exploration. 2) Memory: Records subgraphs, reasoning paths, and sub-objective statuses to provide historical information for reflection. 3) Reflection: Employs LLMs to decide whether to self-correct paths and which entities to backtrack to based on memory.We adapt their code from their GitHub repository[10] to use the KG constructed by REMINDRAG (More details can be seen at Appendix E.2.1).

For all methods described above, the "nomic-ai/nomic-embed-text-v2-moe" model is uniformly used as the embedding model. Methods incorporating knowledge graph memory functionality utilize ChromaDB for knowledge graph storage.

### D.6 Implementation of LLM-as-judgment

Followed previous work [24],We employ the prompt shown in Table 11 to conduct LLM-as-judge (GPT-4o) based answer evaluation.

### D.7 Additional Experiments on More Baseline Models

For better understanding, we consider an additional baselines MemoRAG [36]. This framework enhances RAG to tackle long-context processing, overcoming conventional RAG's reliance on explicit queries and well-structured knowledge. Inspired by human cognition, it uses a dual-system architecture: a light long-range system builds global memory (via configurable KV compression) optimized by RLGF (rewarding clues that boost answer quality), and an expressive system generates final answers. For tasks, the light system produces draft clues to guide retrieving relevant evidence from long contexts, then the expressive system outputs the final answer. It outperforms advanced RAG methods (e.g., GraphRAG [9]) and long-context LLMs in both QA and non-QA tasks, with balanced inference efficiency.

**Experimental Setup**. Given MemoRAG [36]'s capability to manage memory for ultra-long texts, we focused our evaluation on the *LongDependencyQA* task of the LooGLE dataset [24] in this experiment; for the baseline configuration, the model employed is *gpt-4o-mini*, the embedding_model is uniformly set to *nomic-ai/nomic-embed-text-v2-moe* (consistent with our experimental setup), and all other parameters adhere to MemoRAG's default settings—including the mem_mode parameter, which adopts its default value of *memorag-mistral-7b-inst*. The pre-trained weights of *memorag-mistral-7b-inst* were obtained from the Hugging Face Hub, and we verified the model's license

---

[10]https://github.com/liyichen-cly/PoG

agreement (Apache 2.0) on its official Hugging Face Model Card to ensure compliance with research usage requirements, avoiding potential issues related to intellectual property or usage restrictions.

| Methods | Baseline Model | Long-Dependency-Task (%) |
|---|---|---|
| MemoRAG | gpt-4o-mini | 46.97 |
| HippoRAG2 | gpt-4o-mini | 39.60 |
| REMINDRAG | gpt-4o-mini | 57.04 |

Table 5: Performance on Long-Dependency-Task (Accuracy ↑).

**Experimental Results**. As shown in Table 5, MemoRAG indeed demonstrates strong performance in long-dependency tasks: its 46.97% accuracy on the Long-Dependency-Task outperforms HippoRAG2's 39.60% by 7.37 percentage points, a result supported by its dual-system architecture that efficiently builds global memory and optimizes clue selection via RLGF. However, it still lags significantly behind our REMINDRAG method—our approach achieves 57.04% accuracy, a 10.07-percentage-point lead over MemoRAG, as MemoRAG's KV compression may lose fine-grained long-range information, while REMINDRAG better captures dynamic, long-distance associations critical to the task.

# E  Additional Experiment Analysis

## E.1  Additional Analysis on REMINDRAG

**Contextual Information Capture Capability.** We designed ablation experiments to investigate the impact of Contextual Skeleton Network in REMINDRAG's knowledge graph variants on performance, with results shown in Table 6. The experimental results demonstrate that removing the Contextual Skeleton Network leads to a slight performance degradation in REMINDRAG (no effect was observed in Multi-Hop question scenar-

| Methods | Long | Multi-Hop | Simple |
|---|---|---|---|
| REMINDRAG | 57.04% | 74.22% | 76.67% |
| -w/o CS | 51.01% | 74.22% | 73.07% |
| HippoRag2 [13] | 38.06% | 68.04% | 73.08% |

Table 6: Ablation Study (Accuracy ↑). "CS" denotes the Contextual Skeleton Network in the Knowledge Graph. Using "GPT-4o-mini" as the backbone.

ios since the input data inherently consists of discrete single documents without multi-chunk cases within individual documents). Through in-depth analysis, we identify that this performance decline may stem from the following mechanism: The token-based chunking strategy may forcibly segment semantically coherent text, thereby compromising textual integrity. For instance, when the referent entity in a text fragment appears as a pronoun, the knowledge graph typically fails to establish direct associations between this entity and the text fragment. In contrast, large language models (LLMs) can actively access relevant contextual information through the chain connections of the Contextual Skeleton Network, thereby compensating for this deficiency.

**The Impact of Different Hop Count Settings.** We investigate the influence of various hop count configurations on system performance. Specifically, through experiments, we analyze the effect of the Maximum Hop Count parameter in the RE-

| Methods | Avg. API Calls | Token Ratio | Total Tokens |
|---|---|---|---|
| REMINDRAG | 2 | 2.64 | 1.25M |
| GraphRAG [9] | 14.73 | 15.42 | 7.31M |
| HippoRag2 [13] | 2 | 3.91 | 1.87M |

Table 7: Graph Building Comparative Experiment.

MINDRAG model (the relevant parameter definitions are detailed in Appendix D.4). As illustrated in Figure 13, the experimental results demonstrate a significant positive correlation between model performance and hop count settings: as the Maximum Hop Count value increases, the classification accuracy of the system exhibits a monotonically increasing trend. This finding aligns with the

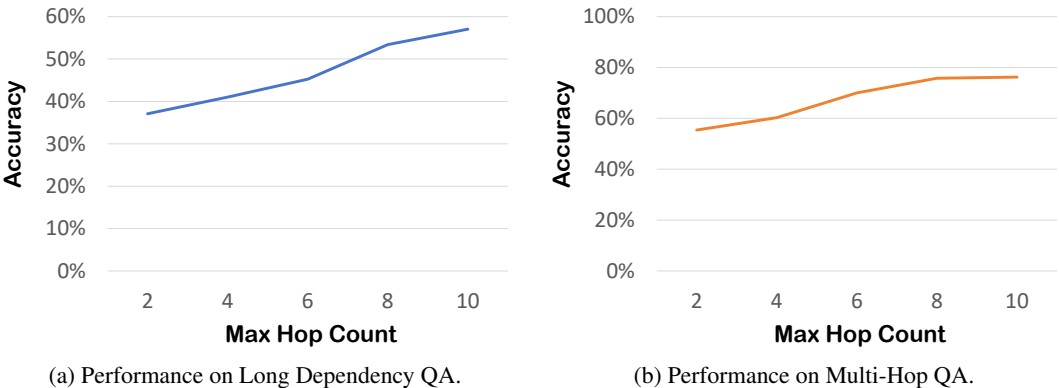

(a) Performance on Long Dependency QA.
(b) Performance on Multi-Hop QA.

Figure 13: The Impact of Different Hop Count Settings.

fundamental principle of the LLM-Guided Knowledge Graph Traversal in REMINDRAG, where a larger hop count enables nodes to access broader neighborhood information, thereby enhancing their information-gathering capability.

**Cost Analysis of Graph Building.** We compare the graph building costs between REMINDRAG and other baselines. As shown in Table 7, experimental results on the LooGLE dataset demonstrate that our method achieves a significantly lower ratio of consumed tokens to source text tokens during the construction process compared to high-overhead systems like GraphRAG[20], while maintaining competitive performance against other baseline approaches.

**Analysis on Path Consistency**. To demonstrate that semantically similar queries correspond to similar graph paths, we design an experiment to quantify the similarity between search paths of an original problem and a similar problem. We model the search paths as sets of *edges*, where an "edge" refers to a direct connection between two nodes. Let $E_1$ represent the set of edges in the original problem's search path, and $E_2$ represent the set of edges in the similar problem's search path.

The similarity $S$ between $E_1$ and $E_2$ is defined as the ratio of the number of identical edges (intersection of the two sets) to the total number of unique edges (union of the two sets after deduplication). Formally, this is expressed as:

$$S = \frac{|E_1 \cap E_2|}{|E_1 \cup E_2|}$$

Here, $|E_1 \cap E_2|$ denotes the cardinality of the intersection (i.e., the count of edges shared by both search paths), and $|E_1 \cup E_2|$ denotes the cardinality of the union (i.e., the total count of distinct edges present in either search path). A $S$ value closer to $1$ indicates a higher overlap between the two search paths (implying more similar underlying logic), while a value closer to $0$ indicates greater divergence.

To obtain a comprehensive measure, we define the aggregated similarity $S_{\text{avg}}$ as the average of all individual similarity values from multiple calculations:

$$S_{\text{avg}} = \frac{1}{n} \sum_{i=1}^{n} S_i$$

where $n$ is the total number of similarity calculations performed, and $S_i$ is the similarity from the $i$-th calculation. We first ran the original problems and then the corresponding similar problems. Taking the base model GPT-4o-mini as an example, the experimental results for similarity are presented in Table 8. From these experimental results, it is evident that semantically similar queries induce similar graph paths.

| Task | $S_{\text{avg}}$ |
|---|---|
| LongDependencyQA | 53.78% |

Table 8: Similarity results for the base model GPT-4o-mini.

**Query Order Impact**. To evaluate whether the order of questions (introduced via random shuffling) affects the system's performance, we designed an experiment where questions in the dataset were randomly shuffled *before each system update*, with three accuracy measurements taken to assess the system's robustness to input order variations.

Specifically, we integrated a random question shuffling module into our codebase. The experiment proceeded in three stages: - For *Update times = 0*: We first randomly shuffled the questions and measured the system's accuracy without any prior updates; - For *Update times = 1*: After the first system update, we re-randomly shuffled the questions (generating a new order distinct from the first) and measured accuracy; - For *Update times = 2*: After the second system update, we again re-randomly shuffled the questions (generating a third unique order) and measured accuracy.

For each stage, we also recorded the *original accuracy* (i.e., accuracy with unshuffled questions) for comparison. The experimental results are summarized in Table 9.

| Update times | Accuracy (With Re-shuffled Questions) | Original Accuracy (Unshuffled) |
|---|---|---|
| 0 | 56.37% | 57.04% |
| 1 | 55.78% | 56.48% |
| 2 | 57.71% | 58.01% |

Table 9: System accuracy on Long-Dependency-Task with re-randomized question orders before each update (with original accuracy for comparison). A new shuffled order was used for each update stage.

From Table 9, we observe that even with a fresh random shuffle of questions before each update, the system's accuracy remains consistently close to the original accuracy (with unshuffled questions) across all stages. Specifically, at Update times = 0, the shuffled accuracy (56.37%) is comparable to the original (57.04%); after the first re-shuffle and update (Update times = 1), the gap remains minimal (55.78% vs. 56.48%); and after the second re-shuffle and update (Update times = 2), the trend persists (57.71% vs. 58.01%). These results confirm that repeated random shuffling of question order—even before each system update—has negligible impact on performance, demonstrating the system's strong robustness to input order variations.

## E.2 Additional Analysis on Baselines

### E.2.1 Analysis of PoG

In the field of KGQA (Knowledge Graph Question Answering), we select the latest method PoG (a self-correcting adaptive planning paradigm for KG-augmented LLMs) as a representative. Since the original method is exclusively designed for the Freebase knowledge graph [2], we modify the source code of PoG (the revised code is open-sourced on GitHub) to adapt it to our own knowledge graphs, allowing traversal of nodes in our constructed knowledge graphs. (As PoG only handles node-level processing, the knowledge graph provided to PoG excludes the chunk component.) We greatly appreciate their open-source contribution.

We retain PoG's implementation methodology and prompts, only altering the output examples in parts of the prompts to adapt to the format; code differences can be viewed at `https://anonymous.4open.science/r/PoG_adapted-C5C0`.

| QA Type | GPT-4o-mini | DeepSeek-V3 |
|---|---|---|
| Long Dependency QA | 30.29K | 25.88K |
| Multi-Hop QA | 5.53K | 6.96K |
| Simple QA | 27.11K | 21.21K |

Table 10: Comparison of the Average Token Consumption of the PoG Method across Different QA Types

The accuracy rates show that the method of conducting Large Language Model (LLM) traversal search solely on the nodes of the knowledge graph, which lacks all textual information, significantly reduces the search accuracy. Compared with our method, PoG's performance on each dataset drops significantly.

Additionally, DeepSeek-V3 exhibits a more pronounced inclination toward a conservative strategy, leading to a higher search depth and greater token consumption compared to GPT-4o-mini. It frequently outputs null answers due to perceived lack of sufficient information, which is also clearly evident in GraphRAG (Appendix E.2.2).

Since the source code of PoG mentions that the maximum search depth is set to 4, in the case of a knowledge graph relying on long-text associations (the revised code we wrote sets the maximum depth to 10, which is consistent with our method), the performance is poor, and the consumption of tokens has significantly increased compared with ours, as shown in Table 10. However, for the simple multi-hop dataset of HotPotQA, which extracts information from Wikipedia and has extremely limited information volume, token consumption is more optimized than ours. We believe that such a simple question-answering dataset cannot well represent the real-world scenarios, and many questions still require complex information extraction. Therefore, the efficiency optimization of searching in complex datasets is of greater application value. For other methods, since they do not involve LLM traversal, comparing token consumption is of little significance.

### E.2.2 Analysis of GraphRAG

Based on the feedback on similar issues from GitHub users, we hereby analyze the reasons for the relatively poor performance of GraphRAG[11].In order to aim for a low hallucination rate, GraphRAG often outputs responses like "More information is needed." For example, Figure 14 shows a question and its generated answer.

**Title —— José Luis Picardo**

| Query | Generated Response |
| --- | --- |
| How many years did Picardo work on Parador projects before the chain's bankruptcy?
A. Fifteen years
B. Twenty-one years
C. Twenty-five years
D. Twenty-nine years" | I don't have access to enough specific data tables or records that would contain information about Picardo's employment timeline with Parador or the dates of Parador's projects and bankruptcy. Without the supporting data, I cannot provide an accurate answer to this question. |
| **Answer**
D. Twenty-nine years | if you can provide the relevant data tables containing:
1. Picardo's employment records with parador
2. dates of parador projects he worked on
3. date of parador's bankruptcy I would be able to analyze this information and give you a precise answer about the duration of his employment. without this data, any number I might provide would be speculative rather than evidence-based. |

Figure 14: Comparative Illustration of Subtle Difference Questions in the Knowledge Graph.

Deepseek-V3 is more inclined to provide conservative outputs compared to GPT-4o-mini (often requesting more information from users instead of providing direct answers), so its accuracy rate has dropped significantly. We have also found that, unlike other methods, GraphRAG is more effective at retrieving for long-dependency question-answering tasks, while its retrieval ability for short-dependency question-answering tasks has declined.

## F  Theoretical Analysis

This section presents a theoretical analysis of the memory capacity of REMINDRAG. We demonstrate that, given a set of queries exhibiting some degree of semantic similarity, our memorization rules in Equation (1) enable edge embeddings to memorize these queries and store sufficient information, thereby enabling effective memory replay. Formally, for any given edge, when the maximum angle $\theta$ between all query embedding pairs within a query set satisfies Equation (4), the edge can effectively memorize the semantic information within that query set.

$$\theta \leq \lim_{d \to \infty} \left[ 2 \arcsin \left( \sqrt{\frac{1}{2}} \sin(\arccos(\lambda)) \right) \right] \tag{4}$$

---

[11]https://github.com/microsoft/graphrag/issues/682

**Assumption 1** *For a typical path $e = A \to B$ in the knowledge graph where $e \in E$, we define that this path "remembers[12]" query $q$ when $q \cdot e > \lambda$. We assume that the input query embedding $q$ is normalized, i.e., $\|q\| = 1$.*

**Proposition 1** *For a set of query embeddings $Q \subseteq \mathbb{R}^n$, we have*

$$\forall q_1, q_2 \in Q, \angle(q_1, q_2) \leq 2 \arcsin \left( \sqrt{\frac{1}{2}} \sin(\arccos(\lambda)) \right) \implies \exists n \in \mathbb{N}, \exists (q_1, q_2, \ldots, q_n) \in Q^n,$$

$$\text{such that} \quad e_{final} = (f_{q_n} \circ \cdots \circ f_{q_1})(\vec{0}) \text{ satisfies } \forall q \in Q, \quad e_{final} \cdot q > \lambda$$

**Proof 1** *Each query embedding $q \in Q$ can be represented as a point on the d-dimensional unit sphere $\mathbb{S}^{d-1}$, i.e., there exists a bijective mapping $\varphi : Q \to N \subseteq \mathbb{S}^{d-1}(0, 1)$ for all $q \in Q$.*

*Assuming $\forall q_1, q_2 \in Q$, their angle satisfies $\angle(q_1, q_2) < \theta$, the corresponding point set $N$ satisfies $\forall p_1, p_2 \in N, d_{\mathbb{S}^{d-1}}(p_1, p_2) < \theta$, where $d_{\mathbb{S}^{d-1}}$ denotes the geodesic distance on $\mathbb{S}^{d-1}$.*

*According to **Jung's Theorem on Spheres** [19, 7] in $\mathbb{S}^{d-1}$, there exists a ball $P = \mathbb{B}^{d-1}(c_p, r_p)$ containing $N$ whose geodesic radius $r_P$ satisfies:*

$$\theta \geq 2 \arcsin \left( \sqrt{\frac{d+1}{2d}} \sin(r_P) \right) \quad \xrightarrow{d \to \infty} \quad \theta \geq 2 \arcsin \left( \sqrt{\frac{1}{2}} \sin(r_P) \right).$$

*Since we typically deal with very high embedding dimensions in practice, we can approximate $d \to \infty$.*

*From Jung's Theorem, the center $c_p$ of the covering ball $P$ lies within the convex hull of $N$. Therefore, the unit vector $c = \varphi^{-1}(c_p)$ satisfies:*

*1) $\forall q \in Q, \angle(c, q) \leq r_P \implies \langle c, q \rangle \geq \cos(r_P)$.*

*2) There exist coefficients $\{\eta_i\}_{i=1}^k \subseteq \mathbb{R}_+$ with $\sum_{i=1}^k \eta_i = 1$ such that $c = \sum_{i=1}^k \eta_i q_i$.*

*From (2), it follows that there exists an update sequence $\{q_{i_j}\}_{j=1}^m \subseteq Q$ such that:*

$$e_{final} = \lim_{m \to \infty} (f_{q_{i_m}} \circ \cdots \circ f_{q_{i_1}})(\mathbf{0}) = c.$$

*Since $f(x) = 2 \arcsin \left( \sqrt{\frac{1}{2}} \sin(r_P) \right)$ is monotonically decreasing:*

$$\lambda \leq \cos(r_P) \implies 2 \arcsin \left( \sqrt{\frac{1}{2}} \sin(\arccos(\lambda)) \right) \geq \theta \implies \forall q \in Q, \langle e_{final}, q \rangle \geq \lambda,$$

*which satisfies the proposition's requirements.*

# G   More Detailed Case Study

## G.1   An Example of a Complete Retrieval Process

In Figure 15, we present a complete example of our retrieval process. This essentially illustrates the detailed procedure of the earlier Case Study in Figure 6.

## G.2   WebUI Screenshots

As shown in Figure 16, we present screenshots of REMINDRAG's web interface, which essentially correspond to the content of the earlier Case Study in Figure 5.

---

[12]We ignore the first term in Equation (2), as it mainly serves to adjust the system's sensitivity to memorized data and is multiplied by a small coefficient, rendering its overall impact on the final result relatively minor.

**Query : Were Scott Derrickson and Conrad Brooks born in the same country?**

---

### Step 1 : Subgraph Initialization

**Get Seed Node :** ['Scott Derrickson', 'Conrad Brooks']

---

### Step 2 : Memory Replay

**Initial Subgraph** :

```
Scott Derrickson -> anchor-2 -> chunk-2
Conrad Brooks
```

Subgraph Info Enough? **False**

---

### Step 3 : Path Expansion Loop   (LLM-Guided KG traversal)

**Loop : 1**

**Add Node :** anchor-2 -> California

Expand Subgraph :

```
Scott Derrickson -> anchor-2 -> chunk-2
                          └-> California
Conrad Brooks
```

Subgraph Info Enough? **False**

**Loop : 2**

**Add Node :** Conrad Brooks -> anchor-9

Expand Subgraph :

```
Scott Derrickson -> anchor-2 -> chunk-2
                          └-> California
Conrad Brooks -> anchor-9
```

Subgraph Info Enough? **False**

**Loop : 3**

**Add Node :** anchor-9 -> chunk-9

Expand Subgraph :

```
Scott Derrickson -> anchor-2 -> chunk-2
                          └-> California
Conrad Brooks -> anchor-9 -> chunk-9
```

Subgraph Info Enough? **True**

---

### Step 4 : Memorizing LLM-Guided KG Traversal

**Filter Path**

Enhance Path :

```
Scott Derrickson -> anchor-2 -> chunk-2
Conrad Brooks -> anchor-9 -> chunk-9
```

Penalize Path :

```
anchor-2 -> California
```

---

### Step 5 : Generate Answer

**Response :** Yes, both Scott Derrickson and Conrad Brooks were born in the United States.

Figure 15: An Example of a Complete Retrieval Process

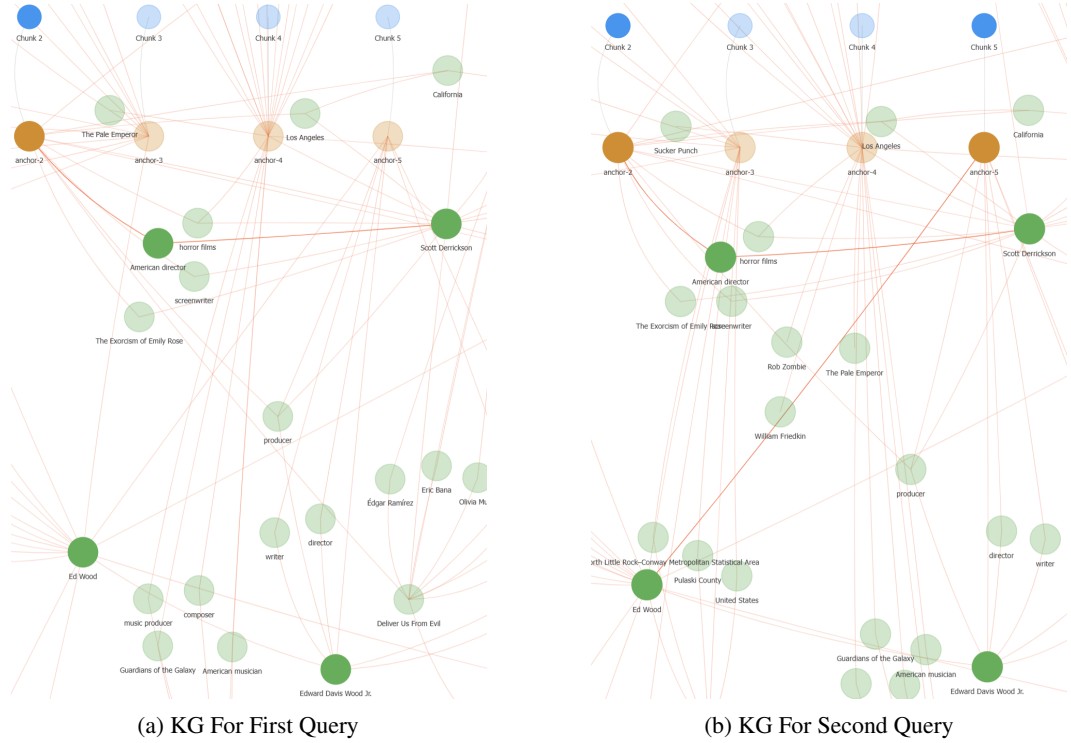

(a) KG For First Query        (b) KG For Second Query

Figure 16: WebUI Screenshots

# H  Broader Impact & Limitations

Currently, large language models (LLMs) face two typical challenges stemming from the Transformer architecture: limited context window capacity and the hallucination problem. Retrieval-augmented techniques offer a viable solution to these critical issues and have gained popularity in various application scenarios. The proposed REMINDRAG achieves high-precision retrieval in diverse complex scenarios, thereby driving comprehensive improvements in human-computer interaction. It enables AI systems to provide more accurate and stable services to society, facilitating the broader integration of AI technologies into daily life.

However, REMINDRAG still exhibits the following limitations: (1) For extremely large documents, constructing a knowledge graph requires substantial time and computational resources; however, it is still much faster than GraphRAG. (2) While the initial retrieval process with LLM-guided knowledge graph traversal leads to significant accuracy improvements, substantial computational resource consumption remains. Our efficiency enhancements focus on subsequent searches; however, further optimization is needed to address the high computational overhead resulting from multiple LLM calls during initialization.

**The Answer Evaluation Prompt**

```
Given one question, there is a groundtruth and a predict answer.
Please decide whether they are the same or not in semantic.
Please only output True or False.
Question: {question}
groundtruth = {reference_answer}
predicted answer = {generated_output}

Only output one word(True or False), without any additional content.
```

Table 11: The Answer Evaluation Prompt

**The Node Selection Prompt**

```
You are a comprehensive analysis expert currently executing a search
in a knowledge graph, evaluating the current search path and
determining the next search node.
The knowledge graph we've constructed is structured as follows:
Entities are divided into two types: one can be called a general **
entity**, and the other, more specific, is referred to as an **anchor
**.
Chunks are connected through anchors to achieve contextual linkage.
Users search by querying general entity nodes until they reach an
anchor node, thereby accessing the corresponding chunks.
Anchors can connect to both general entities and other anchors (and
may also link to chunks), while queryable entities can only connect to
 other queryable entities or anchors.
You are currently at node **{c_node}**. Based on the current search
path and the conditions for determining the next search node, combined
 with the information in the knowledge graph, please provide the most
suitable next search node.
The already traversed path is composed of:
- **{entity_list}**: List of entities visited.
- **{chunk_list}**: List of chunks accessed.
- **{edge_list}**: List of edges traversed.
The adjacency relations of the current node **{c_node}** are:
- **{relation_cnode}**: Relations leading to the next searchable
entity nodes.
- **{connection_cnode}**: Relations leading to the next anchor nodes
connected to chunks.
Your visit history: {c_node_list}, please avoid repeatedly accessing
the same node multiple times.
Additionally, I will provide you with summary information of the
chunks bound to the anchors you have traversed and may potentially
traverse for reference: {anchor_chunk_titles}.
Consider the following two types of information:
1. The semantic relationship between the already queried path, the
summarized **{chunk_summary}**, and the original query **{query}**.
2. The edge weights of the nodes connected to the current node (in **{
relation_cnode}** and **{connection_cnode}**). A higher weight
indicates greater importance.
Determine the most suitable next node (which can be either a
searchable entity node or an anchor node).
If the current node has no unexplored adjacent nodes, then based on
(1), select the most suitable node from the already traversed path (
each node in the path may still have unexplored connections. For
example, in A -> B -> C, if C has no next node, you need to evaluate
(1) and decide whether returning to B is the best choice. In this case
, your task is simply to return B).
Note that you should never select a chunk that has already been
selected.
Anchor nodes are connected to their contextual anchor nodes in the
original document. If you need to access contextual information from a
 section, you can traverse through these anchors.
Please first analyze the above steps, then derive your answer.
At the end of your response, provide the answer in the specified
format-only the node type (either **entity** or **chunk**) followed by
 its ID, without additional explanatory text.
```

Table 12: The Node Selection Prompt

**The Knowledge Graph Filtering Prompt**

```
Now, I have completed a search in a knowledge graph database.

The structure of the constructed knowledge graph is as follows:
Entities are divided into two types: one can be called a general **
entity**, and the other, more specific, is referred to as an **anchor
**.
Chunks are connected through anchors to achieve contextual linkage.
Users search by querying general entity nodes until they reach an
anchor node, thereby accessing the corresponding chunks.
Anchors can connect to both general entities and other anchors (and
may also link to chunks), while queryable entities can only connect to
 other queryable entities or anchors.

Please assist with the following analysis:

Based on the given search path, chunk summaries, and query, analyze
which edges and chunks contain valuable information for answering the
current query.
The current query is **{query}**, and the relationship edges of the
search path are **{edge_list}**, which consists of nodes and chunks.
The chunk summaries are **{chunk_summary}**.

1. **General Judgment Criteria**: Based on the summary text and
previous responses, determine whether they are directly relevant to
the current query.
   - **Relevant Criteria**: Can directly support/refute the conclusion
    of the question or provide key evidence.
   - **Irrelevant Criteria**: Redundant information, off-topic, or
   replaced by more accurate data.
2. **Chunk Relevance Assessment**: Apply the above criteria to each
chunk in the summary.
3. **Edge Relevance Assessment**: Apply the above criteria to each
edge.

The output should include your thought process followed by a JSON-
formatted result. Refer to the example I provided.
Please note that the chunk_id must be a single positive integer only.
For example, "chunk:1" is an incorrect chunk ID, while "1" is correct.
Also, please prioritize selecting valid information from the chunk
whenever possible, as the chunk always contains the most complete
information.

**Output Format**:
(Your thought process in here)
```cot-ans
{{
  "edges": [list of useful edges (using edge IDs)],
   "chunks": [list of useful chunks (using chunk IDs)]
}}
```
```

Table 13: The Knowledge Graph Filtering Prompt

**The Question Transformation Prompt**

```
I'm going to give you a question, the correct answer, and supporting
evidence.
Based on this information, please rewrite the question as a multiple-
choice question with four options.
In the multiple-choice question you create, the correct option should
be {new-ans}.
When creating the incorrect options (distractors), make sure they are
plausible based on the question and evidence, so test-takers can't
easily guess the right answer.
Question: {query}
Correct answer: {ans}
Supporting evidence: {evidence}

When you output the question, provide only the question and its
options (A, B, C, D).
Here's an example of the expected output format:

Example Output:
What is the capital of the United States?
A. New York
B. Washington
C. San Francisco
D. Los Angeles
```

Table 14: The Question Transformation Prompt

**The Similar Questions Rewriting Prompt**

```
Now I'll give you a question, and I want you to rewrite it as a
different question that means the same thing but is phrased
differently.
Original question: {query}

Example:
Original question: Is Beijing the capital of China?
Your Output: Is the capital of the People's Republic of China Beijing?

When you respond, just give me your rewritten question.
```

Table 15: The Similar Questions Rewriting Prompt

**The Subtle Difference Questions Rewriting Prompt**

Question Rewrite

```
Now I'll give you a question, and I want you to rephrase it in a way
that remains as similar as possible to the original but may yield a
different answer.
Please note that your rephrased question must be answerable based on
the reference material.
Original question: {query}
Reference Material: {context}

Example:
Original question: Are both Beijing and Shanghai cities in China?
Your output: Are both Beijing and Chengdu cities in China?

When responding, just provide your rephrased question.
```

Answer Rewrite

```
Now I'll give you a question, and I want you to answer it based on the
 reference material.
Please note that your answer must come from the reference material.
Question: {query}
Reference Material: {context}

Example:
Original question: Are both Beijing and Shanghai cities in China?
Your output: Yes

When responding, just provide your answer.
```

Table 16: The Subtle Difference Questions Rewriting Prompt

**The Option Selection Prompt**

```
Instruction: Given a question and an original answer, please rewrite
the original answer.

If the original answer is not related to any option in the question,
output "I don't know". Otherwise, rewrite the answer to only contain
the actual response to the question without any related analysis or
references.
If the Original answer outputs "I don't know", directly output "I don'
t know".
Please output the rewritten answer directly.

Question = {question}
Original answer = {generated-answer}
```

Table 17: The Option Selection Prompt

**The Entity Extract Prompt**

```
Extract entities and entity relationships from the following text,
then output the complete merged statements of entities and entity
relationships.
An entity should be a simple and easy-to-understand word or phrase.
An entity should be a meaningful word or phrase. For example, in the
sentence "David is seventeen years old," "David" is a meaningful
entity, but "seventeen years old" is not.
An entity is a persistent concept with broad significance, not a
temporary one. For example, "twenty years old," "one hundred dollars,"
 or "building collapse" cannot be entities.
At the same time, an entity typically refers to a specific thing or
concept with a clear identity or definition. For example, in the
sentence "The distance between New York and Boston is not far," "New
York" and "Boston" are entities, but "distance" is not.
When extracting entities, this process should be precise and
deliberate, not arbitrary or careless.
Entity types include organizations, people, geographical locations,
events, objects, professional concepts, etc.
An entity relationship is simply a predicate statement that describes
the subject, object, and their relationship.
Please note that the entities you extract must not include
conjunctions like 'or' or 'and'-they should be precise and standalone.

The output format is as follows:
["entity_1", "entity_2", ..., "entity_n"]

Example 1:
Given text: This travel guide is very detailed, including
introductions to popular attractions, recommendations for local
delicacies, and practical transportation guides.
Output:
["travel guide", "attractions introduction", "food recommendations", "
transportation guide"]

Example 2:
Given text: In this world, police are supposed to catch thieves.
Output:
["police", "thieves"]

Please note: Your final output must strictly follow the required JSON
format and should not include any additional content.
```

Table 18: The Entity Extract Prompt

**The Relation Extract Prompt**

```
For the chunk of text I'm about to input, it contains the following
named entities: {entity_list}.
Please extract the relationships between these named entities. Each
relationship should be a predicate phrase describing the connection
between the subject and the object.
For example, in "Tom" "raises" "dog", "raises" is the relationship.
After extracting a relationship, combine it with the subject and
object to form a complete sentence.

Your final output should be a JSON-formatted list where each sub-list
contains three elements:
[The subject of the relationship, The complete relationship sentence,
The object of the relationship]

I'll provide some examples next for your reference when generating the
 output.

Example 1:
Given text: This travel guide is very detailed, including
introductions to popular attractions, recommendations for local
delicacies, and practical transportation guides.
Output:
[
  ["travel guide", "travel guide includes attractions introduction.",
  "attractions introduction"],
  ["travel guide", "travel guide includes food recommendations.", "
  food recommendations"],
  ["travel guide", "travel guide includes transportation guide.", "
  transportation guide"]
]

Example 2:
Given text: In this world, police are supposed to catch thieves.
Output:
[
  ["police", "police are supposed to catch thieves.", "thieves"]
]

Please note: Your final output must strictly follow the required JSON
format and should not include any additional content.
```

Table 19: The Relation Extract Prompt

