# OpenReview forum: "ReMindRAG: Low-Cost LLM-Guided Knowledge Graph Traversal for Efficient RAG"
_NeurIPS.cc/2025/Conference — NeurIPS 2025 poster_

### Official Review · Reviewer_7sXk · 2025-07-03

**Clarity:** 3
**Significance:** 2
**Originality:** 2
**Rating:** 4
**Confidence:** 3

**Summary:**

The paper introduces KG-RAG, a retrieval-augmented generation system that (i) builds a heterogeneous knowledge graph from documents, (ii) lets an LLM decide which node to expand while answering a query, and (iii) stores the successful traversal as an edge-level memory so that similar future queries can be served with fewer LLM calls. Experiments on three QA benchmarks show higher answer accuracy and lower token consumption than several graph-based RAG baselines.

**Questions:**

### Questions

1.1 Why doesn’t the paper follow the evaluation protocol used in HippoRAG and HippoRAG2, such as reporting Recall@5 and Exact Match? Given that HippoRAG2 is the main competitor, this seems like a natural baseline.
1.2 If the QA datasets have already been transformed into multiple-choice format, why rely on GPT-4o as a judge for evaluation? Why not directly compare the selected answer with the ground truth?
2.Please include an experiment to test how different permutations of question order affect overall performance.
3. please run an experiment to verify whether semantically similar queries in your dataset actually lead to similar graph paths.

### Format issues
- line 486: Figure ??
- figure 11, 12: ORINGIN
- Table 2: Long Denpendency
- Line 47: It efficiently store and recall experience
- “LLM-as-judgment” & “LLM-as-a-judge” used interchangeably, pick one only

**Ethical Concerns:**

["NO or VERY MINOR ethics concerns only"]

**Final Justification:**

In the rebuttal phase, the authors provide clarification and additional experiments that mostly address my concerns. Reflecting these changes, I raised the score from 3 to 4.

**Limitations:**

yes

**Paper Formatting Concerns:**

not that i'm aware of

**Quality:**

2

**Strengths And Weaknesses:**

### Strength
1. The paper is generally well written and has clear motivation. The paper explains the cost–coverage trade-off in graph-based RAG and positions its contribution well.
2. Memory replay is efficient as shown in their experimental section.

### Weakness

1. The only evaluation of effectiveness relies on LLM-as-a-judge, using GPT-4o as the backbone. This is quite problematic, as the evaluation model may share biases with the smaller generation model (GPT-4o mini). Moreover, there are alternative ways to evaluate against ground truth (see Question 1), which the paper does not explore.

2. The paper claims that the results are nearly deterministic, with the only source of randomness being the LLM’s API. However, it overlooks the randomness introduced by the order in which queries enter the system. Since the method depends on remembering the path of previously answered queries, this order can significantly impact outcomes (see Question 2).

3. The memory replay mechanism could become extremely ineffective when semantically similar queries follow entirely different graph paths.

---

> ### Author Rebuttal · Authors · 2025-07-31
>
> Dear Reviewer 7sXk,
>
> Thank you for your detailed feedback and the effort you’ve put into refining your comments on our paper. Regarding the weaknesses in our paper that you have pointed out and your questions, we hereby provide a detailed response to each point, hoping to clarify our work further and address your concerns adequately.
>
> ### Weaknesses
>
> 1. Please refer to Answer 1.2.
> 2. Your point is well-founded, and we have overlooked this issue. However, in actual query processes, the probability that two different questions from our dataset follow the same path is extremely low, so the impact of their order is not significant for our system.
> 3. Please refer to Answer 3.
>
> ### Questions
>
> 1.1 **Why doesn’t the paper follow the evaluation protocol used in HippoRAG and HippoRAG2, such as reporting Recall@5 and Exact Match? Given that HippoRAG2 is the main competitor, this seems like a natural baseline.**
> Due to time constraints, we applied the LooGLE dataset to both REMINDRAG and HippoRAG 2, and the following are the recall results we obtained. (Base model: gpt-4o-mini)
>
> Our recall rate calculation is entirely based on whether the evidence sentences provided by the test dataset are present in the traversed chunks. If all evidence sentences are hit, the recall rate for this sample is 100%.
>
> |                  | HippoRAG2 | REMINDRAG |
> | ---------------- | --------- | --------- |
> | LongDependencyQA | 36.58%    | 60.44%    |
>
>
> We believe that recall rate does not fully reflect the success of a system. In essence, during the LLM-guided traversal and search process, the correct answer can often be inferred without all the evidence. Therefore, we consider accuracy to be a more scientific metric than recall rate for evaluating the effectiveness of a model system.
>
> 1.2  **If the QA datasets have already been transformed into multiple-choice format, why rely on GPT-4o as a judge for evaluation? Why not directly compare the selected answer with the ground truth**
> This is because, to compare with other RAG systems, their outputs may include a large amount of additional information besides the options. Having the large model determine whether the correct answer is included is a very simple task for it (for current large models, such a straightforward semantic coverage judgment task is almost error-free). This simplifies the evaluation process while maintaining fairness.
>
>
> 2.   **Please run an experiment to verify whether semantically similar queries in your dataset actually lead to similar graph paths.**
>
> | Update  times | Accuracy |
> | ------------- | -------- |
> | 0             | 56.37%   |
> | 1             | 55.78%   |
> | 2             | 57.71%   |
>
>
> We greatly appreciate your valuable feedback. We have incorporated the handling of random shuffling of questions in the code and conducted three consecutive experiments, with the results presented in the table above. It can be seen that random shuffling has almost no impact on our system. In this experiment, the questions in each iteration were also randomly shuffled, which indicates that the shuffling of question order likewise has almost no impact on the system's memory capability.
>
> 3. **Please run an experiment to verify whether semantically similar queries in your dataset actually lead to similar graph paths.**
>
> We only use Memory Reply before LLM-Guided Traversal. For similar queries (i.e., those with similar embeddings), they will inevitably point to the same Seed Nodes and, through Memory, obtain the same graph paths. The entire process is derived through purely mathematical methods without involving any LLM. Therefore, in summary, we can theoretically conclude that similar queries point to similar graph paths.
>
> Below is our experiment to demonstrate this point:
>
> To calculate the similarity between the search paths of an original problem and a similar problem, we first model their search paths as sets of "edges" (here, "edges" refer to direct connections between two nodes). Let  $$E_1$$ represent the set of edges in the original problem's search path, and $$E_2$$ represent the set of edges in the similar problem's search path.
>
> The similarity $$ S $$ is defined as the ratio of the total number of identical edges in both searches to the total number of unique edges (after deduplication) across both searches. The formula is:   $$S = \frac{|E_1 \cap E_2|}{|E_1 \cup E_2|}$$
>
> Here,  $$|E_1 \cap E_2|$$ denotes the size of the intersection of the two edge sets (i.e., the number of edges shared by both search paths), and $$|E_1 \cup E_2|$$ denotes the size of the union of the two edge sets (i.e., the total number of distinct edges present in either search path).   A value of \( S \) closer to 1 indicates a higher overlap between the two search paths, meaning their underlying logic is more similar. Conversely, a value closer to 0 indicates greater divergence between the paths.
>
> The final result $$S_{\text{all}}$$ is defined as the average of all similarities, i.e.:
>
> $$S_{\text{all}} = \frac{1}{n} \sum_{i=1}^{n} S_i$$
>
> where $$n$$ is the total number of similarities calculated, and $$S_i$$ is the search path similarity obtained from the i-th calculation.
>
> We first ran the original problems, and on this basis, ran the similar problems; the final results are as follows: (Base model: gpt-4o-mini)
>
> |                  | Similarity ($$S_{\text{all}}$$) |
> | ---------------- | ------------------------------- |
> | LongDependencyQA | 53.78%                          |
>
> From the experimental results, it can be seen that semantically similar queries indeed lead to similar graph paths.
>
>
>
> We hope our responses have effectively addressed your concerns. Going forward, we will release the supplementary code for the experiments conducted in support of this rebuttal as open-source, and further refine our paper in light of your comments.

---

> > ### Comment · Reviewer_7sXk · 2025-08-05
> > **Thank you!**
> >
> > Thanks, the authors, for the detailed rebuttal and additional experiments. Most of my concerns are adequately addressed. In light of the given rebuttals, I raised the score to 4.

---

> > > ### Author Response · Authors · 2025-08-07
> > > **Thank you for your recognition**
> > >
> > > Thank you sincerely for recognizing our work and for your decision to raise the score. Your feedback has been invaluable in helping us refine our work, and we will continue to polish the paper with great care.

---

### Official Review · Reviewer_mkn9 · 2025-07-03

**Clarity:** 4
**Significance:** 3
**Originality:** 3
**Rating:** 5
**Confidence:** 4

**Summary:**

This paper proposes ReMindRAG, a method that simultaneously improves both effectiveness and cost efficiency of KG-RAG systems by adding memory replay functionality. The system employs systematic LLM-guided graph traversal through node exploration and exploitation, along with train-free updates that store LLM-guided traversal experiences in edge embeddings, enabling fast retrieval by loading stored subgraphs through memory replay when similar queries arise. It demonstrated an average of 50% reduction in token consumption and 5-10% performance improvement. Through extensive experiments and detailed experimental procedures, the paper validates the effectiveness and reproducibility of the method.

**Questions:**

1. While the paper presents efficiency analysis primarily in terms of token consumption, it would also be helpful to understand how the proposed method compares to other baselines in terms of latency. Could you provide a comparison of inference latency across different memorization steps or against other methods?
2. Could you provide memory hit rates (percentage of edges retrieved from memorized vs. newly traversed) for each query type?
3. The paper emphasizes the self-correction mechanism, and performance improves as the number of memorization steps increases. What is the proportion of cases where previously incorrect answers are corrected, and cases where previously correct answers become incorrect as memorization steps increase?
4. A missing figure exists. In line 486. Appendix A.2: There is no Figure explaining automated knowledge graph building.

**Ethical Concerns:**

["NO or VERY MINOR ethics concerns only"]

**Final Justification:**

The paper makes a solid technical contribution through its novel memory replay mechanism for KG-RAG systems. The authors have been highly responsive to the reviewers' feedback, providing substantial additional experiments. Therefore, I decided to maintain my current score.

**Limitations:**

yes

**Quality:**

4

**Strengths And Weaknesses:**

### Strengths
1. Cost-Effectiveness Optimization: Across diverse benchmarks the method improves accuracy by 5–10 percentage points while reducing LLM token cost by roughly 50 %, proving superior cost-effectiveness.
2. Training-Free Memory Storage via Memory Replay: Traversal paths are instantly written into edge embeddings, enabling broad reconstruction of the initial sub-graph for later similar or identical queries without invoking the LLM
3. Self-correction capability: Even when an incorrect path is taken, memory re-weighting and penalties correct the mistake, allowing the system to rapidly return to the proper path on subsequent queries.

### Weaknesses
1. The experimental results suggest that this method is both effective and efficient compared to other approaches, and the efficiency aspect is well supported. However, it is not clearly shown whether the method is also more effective than existing LLM-guided knowledge graph traversal approaches in particular, or how it achieves such effectiveness.
2. The experimental evaluation shows promising memory efficiency gains in controlled settings where identical queries are repeated, achieving approximately 50% token savings. However, real-world scenarios typically involve diverse, non-repetitive query patterns. Therefore, it remains questionable whether memory replay would occur sufficiently in practice to provide substantial token savings.

---

> ### Author Rebuttal · Authors · 2025-07-31
>
> Dear Reviewer mkn9,
>
> Thank you for recognizing our work—we truly appreciate it. We also value the time and effort you’ve dedicated to reviewing our paper; your feedback both validates our work and highlights areas needing clarification. Regarding the weaknesses in our paper that you have pointed out and your questions, we hereby provide a detailed response to each point, hoping to clarify our work further and address your concerns adequately.
>
> ### Weaknesses
>
> 1. **Why more effective than baselines**
>
>    Thank you for your feedback. We contribute our effectiveness to the use of KG for LLM inference.
>
>    Unlike methods like PoG and ToG, which introduce LLM traversal for the Freebase database (Freebase itself is a massive knowledge base consisting of entities (nodes) and clearly defined relationships (edges) with a relatively fixed structure), we have not yet found relevant articles similar to ours that introduce LLM traversal for RAG processing of complex texts. We believe their methods have very limited applications because they cannot be separated from the Freebase database, and this approach of fully relying on graphs largely damages the intrinsic semantic information of texts. The heterogeneous knowledge graph we constructed is not an independent database separated from texts, but a dynamic representation tightly coupled with the original texts. Therefore, in our comparisons, we adapted the PoG code to our own text-constructed knowledge graph while fully retaining its original methods, and included it as a comparative model in the paper. This is a compromise made because we did not find methods similar to ours that combine LLM traversal with heterogeneous knowledge graphs formed by text processing.
>
> 2. **How to deal with non-repetitive query.**
>
>    As discussed in our Conclusion section, this challenge can be effectively mitigated by initializing our model with pre-existing, well-curated FAQs relevant to the deployment domain. This approach provides a robust baseline memory, enabling efficient handling of common scenarios and accelerating the graph traversal process.
>
> ### Questions
>
> 1. **Inference latency across different memorization steps**
>
>    Our efficiency is quantified in our paper through token consumption (detailed in Table 2 and Table 3). Our decision not to use and can not use precise wall-clock time latency as a direct metric stems from two key reasons: (1) the strong correlation between LLM inference time and the number of output tokens, and (2) the inherent uncertainties and fluctuations caused by network latency during API calls to LLMs, which would introduce inaccuracies into direct time measurements. Consequently, we kindly refer you to our reported token consumption as the robust and reliable indicator of time cost.
>
> 2. **More analysis on memory hit rate**
>
>    We sincerely apologize that we are unable to complete this additional experiments within the limited timeframe of this rebuttal period. However, we attach great importance to your valuable suggestions and are fully committed to conducting these experiments and incorporating them into revisions
>
> 3. **success rate of self-correction**
>
>    We recorded the questions answered correctly and those answered incorrectly in each experiment. We conducted three experiments on  the LongDenpendencyQA datasets. (i.e., we updated it twice based on the first experiment), and the resulting data are as follows:
>
>    To rigorously evaluate the model's performance changes, we define the following notations: Let the complete set of test cases be $S = \{q_1, q_2, \dots, q_N\}$, where $N = |S|$ is the total number of cases. For the $i$-th experiment ($i \in \{1, 2, 3\}$), the result of each case $q_k$ is denoted by $R_i(q_k)$, where $R_i(q_k) = 1$ for a correct answer and $R_i(q_k) = 0$ for an incorrect answer.
>
>    Based on this, we define the metrics for evaluating changes in consecutive experiments ($i$ vs. $i-1$) as well as the overall change between the third and first experiments ($3$ vs. $1$).
>
>    Proportion of Corrected Cases in Consecutive Experiments:  $$    P_{\text{fix}, i} = \frac{|\{q_k \in S \mid R_{i-1}(q_k) = 0 \land R_i(q_k) = 1\}|}{N}    $$
>
>    Proportion of Regressed Cases in Consecutive Experiments:$$    P_{\text{regress}, i} = \frac{|\{q_k \in S \mid R_{i-1}(q_k) = 1 \land R_i(q_k) = 0\}|}{N}    $$
>
>    Proportion of Corrected Cases (3rd vs. 1st Experiment):$$    P_{\text{fix}, 3 \text{ vs } 1} = \frac{|\{q_k \in S \mid R_1(q_k) = 0 \land R_3(q_k) = 1\}|}{N}    $$
>
>    Proportion of Regressed Cases (3rd vs. 1st Experiment):$$    P_{\text{regress}, 3 \text{ vs } 1} = \frac{|\{q_k \in S \mid R_1(q_k) = 1 \land R_3(q_k) = 0\}|}{N}    $$
>
>    The Net Improvement Rate (P_net) is defined as the "Proportion of Corrected Cases" minus the "Proportion of Regressed Cases", i.e., $P_{net} = P_{fix} - P_{regress}$
>
>    |                        | Fix                                         | Regress                                          | $$P_{net}$$ |
>    | ---------------------- | ------------------------------------------- | ------------------------------------------------ | ----------- |
>    | 1-turn Memorization    | $$P_{\text{fix}, 2}$$ = 3.36%               | $$P_{\text{regress}, 2}$$ = 4.69%                | -1.34%      |
>    | 2-turn Memorization    | $$P_{\text{fix}, 3}$$ = 10.74%              | $$P_{\text{regress}, 3}$$ = 5.37%                | +5.37%      |
>    | 3rd vs. 1st Experiment | $$P_{\text{fix}, 3 \text{ vs } 1}$$ = 8.72% | $$ P_{\text{regress}, 3 \text{ vs } 1}$$ = 5.37% | +4.03%      |
>
>    Due to time constraints, we only conducted experiments on a subset of the LongDenpendencyQA datasets.
>
>
>
> We hope our responses have effectively addressed your concerns. Going forward, we will release the supplementary code for the experiments conducted in support of this rebuttal as open-source, and further refine our paper in light of your comments.

---

> > ### Comment · Reviewer_mkn9 · 2025-08-06
> >
> > I appreciate the authors' detailed rebuttal and thoughtful clarifications. While a few responses could have been more thorough, the overall reply sufficiently addresses the core concerns.
> > Including additional experiments on robustness to query ordering, as raised in Reviewer 7sXk’s second weakness, would further strengthen the work.
> > I believe the authors have responded well to the key points, and I will maintain my current score.

---

> > > ### Author Response · Authors · 2025-08-07
> > >
> > > We are deeply grateful for your detailed feedback and, in particular, your strong recognition of our work—this means a great deal to our team and serves as significant encouragement for our ongoing efforts. We also greatly appreciate your acknowledgment of our rebuttal efforts and your insightful reminder regarding the necessity of thoroughly addressing core concerns.
> > > We have paid close attention to your emphasis on the issue raised by Reviewer 7sXk concerning the robustness to query ordering. Having addressed Reviewer 7sXk's concerns, we present the corresponding experimental results below:
> > >
> > > **Experiment on the impact of question order permutations on overall performance.**
> > >
> > > | Update  times | Accuracy |
> > > | ------------- | -------- |
> > > | 0             | 56.37%   |
> > > | 1             | 55.78%   |
> > > | 2             | 57.71%   |
> > >
> > > Following your valuable feedback, we integrated the handling of random question shuffling into our code and conducted three consecutive experiments, with results summarized in the table above. These results demonstrate that random shuffling of question order has a negligible impact on the performance of our system. Additionally, in each iteration of the experiment, questions were randomly shuffled, which further indicates that such shuffling exerts minimal influence on the system's memory capability.
> > > Thank you again for your constructive input, which has helped strengthen our work.

---

### Official Review · Reviewer_mgBN · 2025-07-03

**Clarity:** 3
**Significance:** 2
**Originality:** 3
**Rating:** 3
**Confidence:** 5

**Summary:**

This paper introduces ReMindRAG, a KG-RAG system designed to improve both performance and cost-efficiency. It employs an LLM-guided traversal strategy for complex information retrieval and introduces a “memory replay” mechanism to reduce costs for subsequent, similar queries. The authors provide a theoretical analysis of the memory's capacity and demonstrate the system's effectiveness and cost savings across three QA tasks against several baselines.

**Questions:**

1) Given the high initial traversal cost, what is the typical "break-even" point for ReMindRAG? How many similar queries are required on average to offset the high initial cost of graph traversal compared to simpler RAG methods or dense retriever? How does this affect its viability in domains with diverse query patterns?
2) The “Damped Update” is designed for stability by making it harder to alter “memories” that have been reinforced. If the initial queries lead to flawed LLM judgements, could this mechanism become prematurely locked in “toxic” memories (incorrect traversal paths), making it difficult for subsequent, correct updates to overcome the initial bias?
3) Could the authors elaborate on how the memory capacity would be impacted by a significantly lower embedding dimension $d$, for example, in a resource-constrained setting?

**Ethical Concerns:**

["NO or VERY MINOR ethics concerns only"]

**Final Justification:**

The author provided a detailed rebuttal and new results. The updated results of the dimensionality test the system's performance in the resource-constrained setting. However, I find that some critical issues regarding the system's practical viability remain unresolved, especially new concerns on complexity surfaced during rebuttal (token / API call cost).

**Limitations:**

Yes, the authors acknowledge the primary limitations in Appendix H. The discussion would be strengthened by a quantitative exploration of the cost-benefit trade-off (per the question above) and a deeper analysis of the system's resilience to its backbone LLM's inherent fallibility.

**Quality:**

2

**Strengths And Weaknesses:**

Strength
- The core idea of creating a persistent, training-free “memory” within the KG’s edge embeddings to improve efficiency is interesting. It tackles a practical challenge in mitigating the high cost of LLM API calls in existing KG-RAG systems.
-  The paper is technically sound and well written. The memory replay part is well motivated and reasoned with theoretical analysis.
- The experimental setup is comprehensive that considering three types of QA tasks with various query scenarios.

Weakness
- The system's primary benefit is for subsequent queries, leaving the high "cold start" cost of the initial traversal as a significant practical barrier, especially in domains with non-repetitive queries.
- The memory update loop relies entirely on the LLM’s own fallible judgment, creating a risk that flawed reasoning or hallucinations could be reinforced over time.
- The claim on “self-correction” capability should be more accurately described as learning between queries rather than correcting an error within a single query’s execution.
- The theoretical analysis for memory capacity relies on an approximation of the embedding dimension $d$ approaching infinity. This simplification obscures the fundamental dependency on dimensionality, which could limit the method's applicability in resource-constrained settings.

---

> ### Author Rebuttal · Authors · 2025-07-31
>
> Dear Reviewer mgBN,
>
> Thank you for your feedback. We address your concern in order as follows. Some concerns are closely related, we will response to them together.
>
>
>
> ### Weaknesses
>
> 1. **Cold start for non-repetitive queries.**
>
>    As discussed in our Conclusion section, this challenge can be effectively mitigated by initializing our model with pre-existing, well-curated FAQs relevant to the deployment domain. This approach provides a robust baseline memory, enabling efficient handling of common scenarios and accelerating the graph traversal process.
>
> 2. **Memory update loop relies entirely on the LLM’s own fallible judgment, creating a risk that flawed reasoning or hallucinations may occur**
>
>    Thank you for your feedback. However, we are quite confused about the precise nature of your concern. We interpret your comment in one possible way and address them below. Please do not hesitate to correct us if our interpretation is inaccurate.
>
>    - **Regarding Memory Updates:** If your concern is that our memory updates might suffer from "flawed reasoning or hallucinations" due to LLM judgment, we wish to clarify that this is not the case. Our memory, stored as weights on the Knowledge Graph's edges, is updated using **LLM embeddings**, not LLM-prompted judgments or generated text. This distinction is crucial, as updates based on numerical embeddings inherently do not introduce the type of flawed reasoning or hallucinations associated with direct LLM text generation.
>    - **Regarding LLM-Guided Traversal and Inference:** If your concern pertains to the potential for inaccurate LLM performance during the graph traversal and subsequent inference, you are correct that this is a possibility. When LLMs are used for navigation and reasoning on the graph, their inherent inaccuracies can indeed impact the overall inference quality. However, it is important to note that this is a **common challenge across all LLM-based KG-RAG methods** and represents an inherent characteristic of current LLM capabilities. Addressing this fundamental limitation of LLMs is beyond the scope of our current research, as our work focuses on the architecture and benefits of the ReMindRAG system itself.
>
> 3. **self-correction capability should be more accurately described as learning between queries rather than correcting an error within a single query’s execution**
>
>    Thank you for your suggestion. We will clarify this in the revision
>
> 4. **Dependency on dimensionality, which could limit the method's applicability in resource-constrained settings**
>
>    Our assumption that the embedding dimension is infinitely large is based on the fact that linear embedding models have very high dimensions. For the function $\sqrt{\frac{d + 1}{2d}}$, when d is greater than around 100 (currently, the embeddings of an LLM often exceed 100 dimensions), it can actually be approximated to $\frac{1}{2}$. This approximation thus holds significant practical utility.
>
> ### Questions
>
> 1. **"break-even" point. How many similar queries are required on average to offset the high initial cost of graph traversal compared to simpler RAG methods or dense retriever? How does this affect its viability in domains with diverse query patterns?**  **
>
>    Our total cost consists of two aspects: KG construction and LLM-guided traversal. As our KG construction method aligns with that of the baselines, the time cost for this phase is consistent across all evaluated methods. Consequently, the critical cost for comparison lies in the LLM-guided traversal. In this regard, our token consumption analysis (detailed in Table 2 and Table 3) provides insights into the 'break-even' point of query costs. Specifically, this break-even point is generally achieved after approximately two queries.
>
> 2. **If the initial queries lead to flawed LLM judgements, could this mechanism become prematurely locked in “toxic” memories**
>
>    No, "Damped Update" is designed to prevent changes in memory content caused by random errors of large models or uncommon queries. Such toxic memories can be solved by our memory Self-Correction process. Specifically, for memory correction, it usually involves several consecutive damped updates. As shown in Figure 4, consecutive corrections typically require at most 2 iterations to correct the memory. We will incorporate this into the revision.
>
> 3. **How the memory capacity would be impacted by a significantly lower embedding dimension**
>
> |      | 1/1    | 1/2    | 1/4    | 1/8    |
> | ---- | ------ | ------ | ------ | ------ |
> | 2    | 58.01% | 57.71% | 55.03% | 53.35% |
> | 1    | 56.48% | 52.34% | 54.02% | 53.02% |
> | 0    | 57.04% | 56.37% | 55.36% | 54.69% |
>
> We performed dimensionality reduction on the embedding model, with specific data as above (The X-axis represents the code truncation ratio; 1/8 means selecting the first eighth of the dimensions, 1/4 means selecting the first quarter, 1/2 means selecting the first half, and the Y-axis represents the number of updates. The data in the table represent the answer accuracy rate.) It can be seen that the impact on the entire system is negligible. We speculate that this may be due to the Matryoshka Representation Learning technology it applies, which enables effective dimension truncation [1].
>
> We hope our responses have effectively addressed your concerns. Going forward, we will release the supplementary code for the experiments conducted in support of this rebuttal as open-source, and further refine our paper in light of your comments.
>
>
>
> **Reference**
>
> [1] Kusupati, Aditya, et al. "Matryoshka representation learning." *Advances in Neural Information Processing Systems* 35 (2022): 30233-30249.

---

> > ### Comment · Reviewer_mgBN · 2025-08-05
> >
> > Thank you for the detailed rebuttal and for providing new results. However, I find that the most critical issues regarding the system's practical viability remain unresolved.
> >
> > The primary concerns are:
> > - Reliance on Fallible LLM Judgment: Your response misunderstands the core of this concern. The issue is not the format of the memory update, but that the system's feedback loop can memorize and reinforce the LLM's own flawed reasoning at every stage, including seed node selection, KG traversal, and final judgment (as Reviwer 7sXk pointed out). Although "Damped Update" may partially mitigate it at one stage, dismissing this as a general problem overlooks the unique risk of a system that explicitly uses "memory replay," which can amplify and entrench the LLM's initial errors.
> >
> > - Unsubstantiated Claims on Cost, Recovery, and Sensitivity: The practical claims about the system's efficiency and resilience are not convincingly supported by the evidence provided.
> >
> >   - (Cost, W1/Q1) The assertion of a two-query "break-even" point does not appear to hold up against the results in Table 2. A fair comparison requires amortizing the token costs from the initialization and subsequent runs (multi-turn case); without this, the claim is misleading.
> >
> >   - (Recovery, Q2) The claim that "toxic" memories can be corrected in "at most 2 iterations" is a misinterpretation of Figure 4, which does not provide evidence for the dynamics of multi-step error correction.
> >
> >   - (Sensitivity, Q3) While I appreciate the new results on dimensionality, the conclusion that the impact is "negligible" is premature. The full embedding dimension is not stated, but assuming the cited "nomic-ai/nomic-embed-text-v2-moe" model ($d=768$), the most aggressive reduction tested (⅛ dimension) results in $d=96$. This dimension is still large enough to comfortably satisfy the theoretical requirements of Eq. (4), so this experiment does not adequately test the system's performance in a truly resource-constrained, low-dimensional setting.

---

> > > ### Author Response · Authors · 2025-08-07
> > >
> > > Thank you for your feedback. We are now addressing your remaining concerns sequentially.
> > >
> > > # Reliance on Fallible LLM Judgment
> > >
> > > First, we would like to clarify that the selection of seed nodes in our system is independent of edge vectors. This implies that updates resulting from the LLM-Guided Traversal do not influence the seed node selection process. The LLM solely affects path determination during traversal and assesses whether the currently gathered information is sufficient to answer the user's query.
> > >
> > > Regarding instances of LLM errors, we categorize them into two distinct scenarios for discussion:
> > >
> > > 1. **The LLM makes a flawed inference, but these instances are infrequent, and our system includes mechanisms that allow the LLM to self-correct these errors.** As may be observed in the supplementary experiments provided to Reviewer mkn9, the LLM can quickly recognize and correct these errors in subsequent steps. We have also elucidated this point at the end of Section 3.2.2. Following a "Memory Reply," the LLM evaluates whether it can answer the query. If the information is insufficient, it re-initiates the traversal process, triggering the "Enhance" and "Penalize" memory mechanisms.
> > > 2. **The LLM produces a flawed inference due to its inherent limitations, such as unavoidable hallucinations, constrained reasoning abilities, or biases.** Addressing these core limitations of the LLM itself is outside the scope of our current work. One can conceptualize the LLM as a librarian within a library. The "Damped Update" mechanism provides it with opportunities for trial and error. After the "Guided Traversal," we further task the LLM with a self-assessment to determine what information to "Enhance" and what to "Penalize." The subsequent check after a "Memory Reply" offers another chance to detect errors. Our system is designed to maximize the proactive reasoning capabilities of LLMs in information retrieval; however, it cannot overcome the intrinsic limitations of the LLM itself. Nevertheless, it is foreseeable that as the capabilities of LLMs advance, the prevalence of the first scenario relative to the second will increase, leading to a corresponding improvement in the overall performance of our system.
> > >
> > > # About "break-even point"
> > >
> > > Thank you for seeking clarification on this point. We apologize for any ambiguity in our previous explanation regarding the "break-even point." We previously interpreted the term to mean the point at which overall token consumption stabilizes.
> > >
> > > **To clarify**: Our method fundamentally does not involve a token consumption trade-off between graph construction and traversal. Specifically, our graph construction phase is consistent with existing approaches and introduces no additional token overhead. Consequently, the token efficiencies achieved during graph traversal are a net gain, and there is no "break-even point" where these savings are offset by higher construction costs.
> > >
> > > For a more granular analysis, the token consumption of our method across the utilized datasets is detailed below.
> > >
> > > For the purpose of this discussion, we exclude the average essential per-run costs of the system (e.g., answer generation), which are detailed in the following table.
> > >
> > > | Long Dependency | Multi-Hop QA |
> > > | --------------- | ------------ |
> > > | 2.7k            | 1.9k         |
> > >
> > > Beyond these, the revised Table 2 now exclusively presents the token consumption for **graph traversal**, presented as follows:
> > >
> > > | Methods             | Long Dependency QA (Same Query) | Long Dependency QA (Similar Query) | Multi-Hop QA (Same Query ) | Multi-Hop QA (Similar Query) | Multi-Hop QA (Different Query) |
> > > | ------------------- | ------------------------------- | ---------------------------------- | -------------------------- | ---------------------------- | ------------------------------ |
> > > | 3-turn Memorization | 4.01K                           | 4.32K                              | 3.99K                      | 4.12K                        | 7.86K                          |
> > > | 2-turn Memorization | 4.85K                           | 7.15K                              | 4.66K                      | 7.44K                        | 7.60K                          |
> > > | 1-turn Memorization | 6.98K                           | 11.28K                             | 5.83K                      | 7.98K                        | 8.67K                          |
> > > | No Memorization     | 12.21K                          | /                                  | 8.26K                      | /                            | /                              |
> > >
> > > We acknowledge that our previous statement about a "break-even point" may have been imprecise. It was based on our interpretation of the term as the point where token consumption stabilizes. Under that interpretation, we estimated it would occur at approximately two queries for "Same Query" and three for "Similar Query."

---

> ### Author Response · Authors · 2025-08-07
>
> # Clarification on the Correction of "Toxic Memories"
>
> We apologize for the previous lack of clarity on this topic. To facilitate your understanding, we will now provide a brief quantitative analysis.
>
> Building upon the preceding section, "Reliance on Fallible LLM Judgment," we posit that correctable "toxic memories" are primarily those that arise from low-probability, erroneous inferences made by the LLM. To model this, we introduce a simplification by assuming the query vector and the edge vector are co-directional. (In practice, if they were not co-directional, the penalty coefficient δ(x) would be even larger, strengthening our conclusion). Consequently, the "Damped Update" equations can be simplified as follows:
>
> *(We ask for your understanding regarding our inability to upload images or links, which is a constraint imposed by NeurIPS submission policies. You may later independently verify these functions using tools such as Python or MATLAB.)*
>
> Enhance Function:  $E(x) = x + \frac{2}{\pi} \cos\left( \frac{\pi x}{2} \right)$
>
> Penalize Function:  $P(x) = x - \frac{2}{\pi} \cos\left( \frac{\pi x}{2} \right)$
>
> We then construct three scenarios to simulate the correction of an erroneous enhancement:
>
> 1. **One Enhance, one Penalize:** $y= P(E(x))$
> 2. **One Enhance, two Penalizes:** $y= P(E(E(x)))$
> 3. **One Enhance, three Penalizes:** $y= P(E(E(E(x))))$
>
> We analyze the intersection points of these three scenarios with the line $ y=0.55 $, which represents our chosen confidence threshold λ. The corresponding initial values (x-coordinates) for these intersections are 0.155, 0.336, and 0.476, respectively.
>
> This analysis demonstrates that for an edge whose initial confidence value x is below the threshold λ, even if it is incorrectly enhanced once, its confidence can be restored to a value below λ within at most two subsequent penalization steps in the vast majority of cases (i.e., for initial values up to 0.336).
>
>
>
> # Additional Dimensionality Reduction Experiments
>
> For your reference, we have conducted additional experiments on dimensionality reduction. The results show that our method's performance remains robust even in lower-dimensional spaces. We consider this a practical lower bound for analysis, as real-world applications rarely involve fewer dimensions.
>
> We performed dimensionality reduction on the embedding model to test its robustness. The table below shows the specific data from this experiment. The **X-axis** represents the dimension truncation ratio (e.g., 1/2 means selecting the first half of the dimensions), and the **Y-axis** represents the number of updates. The data in the table is the answer accuracy rate (All experiments use the same default parameters).
>
> |      | 1      | 1/2    | 1/4    | 1/8    | 1/16   | 1/32   | 1/64   |
> | ---- | ------ | ------ | ------ | ------ | ------ | ------ | ------ |
> | 2    | 58.01% | 57.71% | 55.03% | 53.35% | 53.28% | 53.42% | 53.15% |
> | 1    | 56.48% | 52.34% | 54.03% | 53.02% | 52.97% | 52.95% | 52.55% |
> | 0    | 57.04% | 56.37% | 55.36% | 54.69% | 54.36% | 52.78% | 52.08% |
>
> Regarding the use of $ d\rightarrow \infty $ in our theoretical analysis, its purpose is to establish a reasonable range for setting the threshold $ \lambda $. A suitable λ can still be selected even when the dimensionality is not approaching infinity.

---

> > ### Author Response · Authors · 2025-08-09
> > **Thanks for your time!**
> >
> > Dear Reviewer mgBN,
> >
> > As the discussion deadline is closing, we wonder if you have any other concerns. We are glad to engage in further discussion.
> >
> > All authors

---

### Official Review · Reviewer_fchT · 2025-07-04

**Clarity:** 3
**Significance:** 3
**Originality:** 3
**Rating:** 4
**Confidence:** 4

**Summary:**

This paper proposes ReMindRAG, a Retrieval-Augmented Generation (RAG) framework. ReMindRAG employs LLM-guided traversal of document-level knowledge graphs, then stores successful and failed traversal paths within edge embeddings. These embeddings are updated using closed-form rules ("Fast Wakeup & Damped Update") to enable future similar queries to be answered with fewer LLM calls. Experiments across three QA settings (long-dependency, multi-hop, simple) demonstrate that ReMindRAG achieves higher accuracy compared to a set of strong baselines while reducing token costs by approximately 50%. By storing traversal experiences in edge embeddings mimicking LLM parametric memory, it reduces LLM overhead by 50% for recurring queries while improving accuracy by 5–12%.

**Questions:**

- ReMindRAG reduces token usage by about 50%, but is compared to PoG without memory replay (Table 2). Can you provide more direct comparisons with the original PoG configuration?

**Ethical Concerns:**

["NO or VERY MINOR ethics concerns only"]

**Final Justification:**

The author made changes to the questions I raised, and I increased my rating.

**Limitations:**

Please refer to Strengths And Weaknesses.

**Paper Formatting Concerns:**

There are no obvious issues with the format.

**Quality:**

3

**Strengths And Weaknesses:**

Strengths:
- ReMindRAG introduces an innovative combination of LLM-guided graph traversal with node exploration, node exploitation, and memory replay. This approach aims to improve both effectiveness and cost efficiency in Knowledge Graph Retrieval-Augmented Generation (KG-RAG) systems.

- By targeting reduced LLM invocation costs and inference time, ReMindRAG addresses a practical concern for deploying KG-RAG systems at scale, making it relevant for real-world applications.

- The authors demonstrated the effectiveness of ReMindRAG through experiments and theoretical analysis.

Weaknesses:
- Section 4 refers to “Equation (4)” as the storage condition, but the last numbered formula in the section is (3); there is no (4)?

- There are still a lot of errors in the paper, which makes it difficult to read (there are quite a few). For example, in the appendix there is: "illustrated in Figure ??".

- Graph construction still calls LLM for NER and relation extraction; Table 5 shows token counts but not latency. A better comparison on speed can be done with other methods.

- The paper alternates between "ReMindRAG" and "REMINDRAG," suggesting a lack of consistency in naming the proposed method.

- Terms like "memory replay" and "node exploitation" are introduced without clear definitions in the provided text, causing confusion.

- Omits comparison with parametric memory-augmented RAG (e.g., MemoRAG, HippoRAG).

- Table 1 reports the accuracy of Plan-on-Graph (PoG), but does not explain how PoG is adapted to the KG structure of ReMindRAG (Appendix E.2.1). If PoG's original optimization for Freebase is lost, it may lead to an unfair comparison.

---

> ### Author Rebuttal · Authors · 2025-07-31
>
> Dear Reviewer fchT,
>
> Thank you for your feedback. **We will proofread the paper to improve our writing in the revision. Beyond the writing aspects, we address your remaining concerns in the following**. Some concerns are closely related, we will response to them together.
>
> ### Weaknesses
>
> **3. Table 5 shows token counts but not latency. A better comparison on speed can be done with other methods.**
>
> Both the KG construction process and the subsequent LLM-Guided Traversal need to be completed using LLM, which involves time cost. However, since our KG construction method aligns with that of the baselines, the time cost associated with this phase is consistent across all evaluated methods. Therefore, the critical time cost for comparison lies in the LLM-Guided Traversal, which is quantified in our paper through token consumption (detailed in Table 2 and Table 3). Our decision not to use precise wall-clock time as a direct metric stems from two key reasons: (1) the strong correlation between LLM inference time and the number of output tokens, and (2) the inherent uncertainties and fluctuations caused by network latency during API calls to LLMs, which would introduce inaccuracies into direct time measurements. Consequently, we kindly refer you to our reported token consumption as the robust and reliable indicator of time cost.
>
> **5.Definitions of "memory replay" and "node exploitation"**
>
> Our definition of 'memory replay' aligns with its standard usage in the AI field [1-2]. It refers to the recall of valuable past memories to facilitate current tasks. However, our approach diverges from prior work that typically involves recalling explicit samples. Instead, our 'memory' is embedded as weights within the KG, and it is these weights that are exclusively utilized to guide LLM inference and graph traversal. Additionally,  we have provided detailed explanations of node exploitation in Section 3.1 (Line 122-127), and we will further clarify them in the revision.
>
> **6.Comparison with parametric memory-augmented RAG (e.g., MemoRAG, HippoRAG)**
>
> Our baselines are selected to represent a diverse set of KG-RAG methods, including the reviewer-suggested ones, though it's impractical to evaluate every single approach. For example, our experiments already incorporate Hippo2, the upgraded version of Hippo. Note that Memo, which is not a KG-RAG method, was initially excluded. However, at your suggestion, we still performed additional experiments including 'MemoRAG'
>
> - **Setup**. Due to MemoRAG's characteristic of handling memory for ultra-long texts and time constraints, we focused on evaluating LongDependencyQA of the LooGLE dataset here. The baseline model uses gpt-4o-mini, the embedding_model is uniformly adopted as nomic-ai/nomic-embed-text-v2-moe with ours, and other parameters are all MemoRAG's default parameters, including mem_mode using its default memorag-mistral-7b-inst. We downloaded its pre-trained weight files from Hugging Face.
> - **Results**. The results in the following table still show our superiority
>
> | Model           | Baseline Model | Evaluation Task      | Result |
> | --------------- | -------------- | -------------------- | ------ |
> | MemoRAG         | gpt-4o-mini    | Long-Dependency-Task | 46.97% |
> | HippoRAG2       | gpt-4o-mini    | Long-Dependency-Task | 39.60% |
> | REMINDRAG(Ours) | gpt-4o-mini    | Long-Dependency-Task | 57.04% |
>
> **7. Implementation details of PoG**
>
> In Appendix D.4 (Lines 673-682), we detail our use of PoG and have attached the corresponding code. While a complete re-description of PoG's intricate implementation details is omitted from the Appendix, our provided code clearly demonstrates that we have fully retained all core methodologies of PoG, including its two-stage LLM-based relation pruning, LLM-based entity pruning, iterative graph traversal and reasoning, and memory updates during reasoning. Only the processing part for the specific format and rules of the Freebase dataset was removed because it is incompatible with our dataset format. Additionally, the interaction method with the knowledge graph was modified: SPARQL was replaced by an in-memory index. Our knowledge graph is much smaller than the ultra-large Freebase dataset, so in-memory lookup optimization can instead improve PoG's performance. Therefore, we have retained all of PoG's optimization schemes in terms of both accuracy and token consumption.
>
> ### Question
>
> **Q1. Comparisons with the original PoG configuration**
>
> As mentioned earlier, our modified PoG method has been fully open-sourced (available at the link in Section E.2.1 of the paper). It can be seen from the code that we have not modified any of PoG's methods; the changes we made are as explained in point 7 above. PoG itself does not have our memory replay function, which is one of our innovations. You may have confused it with PoG's proposed "Memory Updating": *The information stored in memory provides historical retrieval and reasoning information for reflection. After a two-step exploration, we dynamically update the searched subgraph GSub, reasoning paths P, and sub-objective status S in memory based on the ongoing reasoning process* (Quoted from the original PoG paper) . Its memory mechanism is only for a single retrieval. The original PoG configuration relies on knowledge graphs composed solely of nodes (e.g., Freebase). In terms of heterogeneous graphs, it lacks chunk processing capability, making it more suitable for simple question-answering tasks but less effective in scenarios relying on contextual text processing (in practice, it is difficult to fully represent a piece of text in the form of a node graph, which severely damages the intrinsic semantic information).
>
>
>
> We hope our responses have effectively addressed your concerns. Going forward, we will release the supplementary code for the experiments conducted in support of this rebuttal as open-source, and further refine our paper in light of your comments.
>
> **Reference**
>
> 1. The Effects of Memory Replay in Reinforcement Learning
> 2. MEMORY REPLAY WITH DATA COMPRESSION FOR CONTINUAL LEARNING

---

> > ### Comment · Reviewer_fchT · 2025-08-05
> > **Thanks to the author for the reply, I increased the rating.**
> >
> > Thanks to the author for the reply, I increased the rating.

---

> > > ### Author Response · Authors · 2025-08-05
> > > **Thank you for your recognition**
> > >
> > > Thank you for your feedback. Your comments have helped us improve our work significantly. We greatly appreciate your recognition of our revisions and the adjusted score. Please let us know if you have any further questions, and we will respond promptly.

---

### Official Review · Reviewer_ntvy · 2025-07-05

**Clarity:** 3
**Significance:** 2
**Originality:** 3
**Rating:** 4
**Confidence:** 3

**Summary:**

The paper introduces ReMindRAG, a novel LLM-guided knowledge graph traversal approach aimed at enhancing the efficiency and accuracy of RAG systems. By using techniques such as node exploration, node exploitation, and memory replay, ReMindRAG improves both the effectiveness of the system and its cost-efficiency. The system constructs a knowledge graph from unstructured text, guiding the traversal process with an LLM to progressively explore and exploit nodes for precise answer retrieval. Memory replay is employed to store and recall traversal experiences, reducing reliance on repeated LLM calls and thus cutting down on the cost. The paper demonstrates that ReMindRAG outperforms existing methods, achieving notable improvements in accuracy while reducing the token consumption by about 50%.

**Questions:**

1. Could the authors explore the scalability of ReMindRAG in environments where knowledge graphs are constantly evolving or highly dynamic? How does the system handle real-time updates to the graph?

2. Could the author compare the method's time cost with other RAG methods, which will be beneficial to explain the efficiency of the proposed method.

**Ethical Concerns:**

["NO or VERY MINOR ethics concerns only"]

**Limitations:**

Yes

**Quality:**

3

**Strengths And Weaknesses:**

**Strengths:**

1. The use of memory replay for efficient graph traversal is a novel and highly effective approach. It offers a significant reduction in LLM costs while maintaining high retrieval accuracy.

2. The memory replay mechanism significantly reduces token consumption, especially for repeated queries, demonstrating strong cost-efficiency without sacrificing performance.

3. The paper presents a comprehensive experimental setup, including multiple benchmark datasets and LLM backbones, which solidly demonstrates the effectiveness of ReMindRAG over various baselines.

4. The explanation of the LLM-guided traversal, along with the theoretical and practical aspects of memory replay, is well-articulated and detailed, aiding in understanding the approach.

**Weaknesses:**

1. While the method reduces token consumption in certain scenarios, further evaluation is needed to assess how well the system scales with very large or noisy datasets and in real-time applications.

2. The overall system, while efficient, involves several intricate mechanisms (e.g., memory replay, node exploration/exploitation), which may introduce complexity in real-world applications or during deployment.

3. My main concern of the usefulness of this method is the reliance on high-quality KGs, which limits the value of the method in real-world scenarios.

---

> ### Author Rebuttal · Authors · 2025-07-31
>
> Dear Reviewer ntvy,
>
> Thank you for your insightful comments and the valuable perspectives you’ve shared. Regarding the weaknesses in our paper that you have pointed out and your questions, we hereby provide a detailed response to each point, hoping to clarify our work further and address your concerns adequately.
>
> ### **Weaknesses**
>
> 1. **how well the system scales with very large or noisy datasets**
>
>    In fact, the LooGLE Dataset utilized for our evaluation is already of considerable size. However, addressing noisy datasets and KGs, though a valuable direction, lies beyond the current research scope of our paper and the design of our baselines. While we fully respect your concern, we believe that existing KG-RAG methods are broadly susceptible to noise unless specifically engineered for robustness against it, a focus not within our current work. Additionally, our KG construction method follows established practices (Line 92-102 and Appendix A) and does not introduce any unique steps that would make it more susceptible to noise.
>
> 2. **Complexity in deployment.**
>
>    Our open-source code repository has well-encapsulated our system, allowing you to build a ReMindRAG system with just a few lines of code. Additionally, we support high customizability: you can use your own components such as LLM model, Embedding model, and Chunk model by simply inheriting from the abstract classes in our library. For more details, please refer to the README.md of our open-source repository.
>
> 3. **reliance on high-quality KGs**
>
>    Our Knowledge Graph (KG) is constructed automatically from unstructured continuous text, with the methodology further elaborated in Appendix A. For this purpose, we leverage the Large Language Model. This approach facilitates a convenient construction process that guarantees a high standard of quality without requiring manual oversight or optimization. We concur with your concern that models reliant on specific, high-quality knowledge graphs may lack generalizability. The comparative PoG method we reference is a case in point, as it is restricted to a certain knowledge graph (Freebase). In contrast, our proposed model is designed to overcome this very drawback.
>
>    Many existing methods necessitate high-quality source texts or high-performance LLMs to construct high-quality KGs; however, investigating the construction of high-quality KGs falls outside the scope of our current work.
>
> ### **Questions**
>
> 1. **Evaluation on evolving or highly dynamic KG**
>
>    We would like to highlight that we and our baselines are not designed for noisy KGs or evolving KGs, which are not our research task and scope. However, to address your concern, it's conceptually possible to add new content to the KG at any time without affecting existing data. Such newly added content could undergo automated operations, like integration with synonyms, to be properly incorporated. Crucially, during LLM inference, we ensure the KG remains static (i.e., it is treated as a non-evolving KG). In such scenarios, when newly added nodes or edges are not equipped with memory/weights, the LLM's inference for those specific elements would then depend on its own capabilities.
>
> 2. **Could the author compare the method's time cost with other RAG methods, which will be beneficial to explain the efficiency of the proposed method.**
>
>    Both the KG construction process and the subsequent LLM-Guided Traversal need to be completed using LLM, which involves time cost. However, since our KG construction method aligns with that of the baselines, the time cost associated with this phase is consistent across all evaluated methods. Therefore, the critical time cost for comparison lies in the LLM-Guided Traversal, which is quantified in our paper through token consumption (detailed in Table 2 and Table 3). Our decision not to use precise wall-clock time as a direct metric stems from two key reasons: (1) the strong correlation between LLM inference time and the number of output tokens, and (2) the inherent uncertainties and fluctuations caused by network latency during API calls to LLMs, which would introduce inaccuracies into direct time measurements. Consequently, we kindly refer you to our reported token consumption as the robust and reliable indicator of time cost.
>
>
>
> We hope our responses have effectively addressed your concerns. Going forward, we will further refine our paper in light of your comments.

---

> > ### Comment · Reviewer_ntvy · 2025-08-04
> >
> > Thank you for your detailed reply. Please consider strengthening the paper’s organization, and experimental comparisons in the future version. I will keep my current score in this round.

---

> > > ### Author Response · Authors · 2025-08-04
> > >
> > > Thank you for your valuable feedback. We will strengthen the paper's organization and expand experimental comparisons in the revised version as you suggested.

---

### Note · Authors · 2025-08-15

Dear Area Chair,

Thank you for overseeing the review process. **We are pleased to note all initially negative reviews have been revised to positive after our comprehensive rebuttal,** with detailed responses to reviewers’ concerns. No major issues or fundamental revisions were raised.

**Our strengths acknowledged by the reviewers are as follows**:

- **Important question**: We address the critical challenge of optimizing synergy between KG-RAG system effectiveness and cost efficiency, with strong practical value.
- **Novel method**: Our method integrates LLM-Guided Traversal and a *Memory Replay* mechanism into KG traversal, reducing LLM token consumption by ~50% while maintaining high retrieval accuracy.
- **Empirical analysis**: Comprehensive experiments (spanning multiple benchmarks and backbone LLMs) robustly demonstrate our method’s effectiveness against existing baselines.
- **Theoretical analysis**: Clear theoretical explanations for our method’s efficacy.

**The reviewers’ concerns and our responses are as follows**:

- **Time-cost comparison**: We noted network latency fluctuations cause unstable measurements of time-cost, and clarified our token consumption (strongly correlated with time) is a more reliable proxy.
- **Reliance in memory updates & "toxic memory" risk**: We theoretically and experimentally showed that our "damping update" and Self-correction can mitigate this.
- **Impact of embedding dimensions on memory capacity**: We theoretically and experimentally demonstrated the robustness of the memory capacity across different truncated dimensions
- **More Evaluation & baselines**: We added Recall comparisons to further strengthen our evaluation, and included suggested baselines, which is a non-typical KG-RAG method, for thoroughness.
- **Query order impact & path consistency**: Shuffled query experiments showed robustness; path similarity calculations verified consistency for semantically similar queries.
- **Break-even point**: No typical "break-even point" exists (our KG construction matches prior work, no extra costs). We provide more detailed token consumption analysis to show our stable low costs.

We appreciate the reviewers’ recognition and valuable feedback. Given the post-rebuttal score improvement, positive reception, and our thorough responses, we are confident the paper contributes meaningfully to the KG-RAG community—offering a novel solution balancing accuracy and cost-effectiveness.

---

### Decision · Program_Chairs · 2025-09-17

**Decision:**

Accept (poster)

**Comment:**

This paper proposes ReMindRAG, a knowledge graph–based RAG framework that enhances both accuracy and cost-efficiency through LLM-guided graph traversal and memory replay. It explores and exploits nodes to retrieve precise answers, while storing traversal experiences in edge embeddings to reduce repeated LLM calls. This train-free update mechanism, inspired by parametric memory, enables fast reuse of successful paths for similar queries. Across multiple QA tasks, ReMindRAG achieves 5–12% accuracy gains and reduces token consumption by about 50%, demonstrating consistent improvements over strong baselines.

Strengths:
1. Most reviewers find the integration of LLM-Guided Traversal and a Memory Replay mechanism novel and interesting.
2. This paper has provided theoretical justifications
3. Comprehensive experiments have been provided to demonstrate the effectiveness of the method.

Weakness:
Some questions have been raised regarding the complexity of the proposed method, the faithfulness of the memory update loop and evaluation details.

During the rebuttal, the authors provided:
1. Additional evaluations to demonstrate the method’s effectiveness and robustness, including new baselines, shuffled query experiments, , path similarity analyses for consistency verification, and reliance in memory updates & "toxic memory" risk.
2. Clarification on the evaluation details.

Most reviewers agree that the authors have addressed their concerns and have given the paper positive scores, with only one raising questions about the method’s practical viability. I consider this a minor issue in light of the overall contributions, and therefore recommend acceptance.